# Muscle cell-type diversification is driven by bHLH transcription factor expansion and extensive effector gene duplications

Alison G. Cole [1,2,6] ✉, Stefan M. Jahnel [1,3,6], Sabrina Kaul[1], Julia Steger[1], Julia Hagauer [1], Andreas Denner[1], Patricio Ferrer Murguia[1], Elisabeth Taudes[1], Bob Zimmermann [1], Robert Reischl[1], Patrick R. H. Steinmetz [1,4] & Ulrich Technau [1,2,5] ✉

Animals are typically composed of hundreds of different cell types, yet mechanisms underlying the emergence of new cell types remain unclear. Here we address the origin and diversification of muscle cells in the non-bilaterian, diploblastic sea anemone *Nematostella vectensis*. We discern two fast and two slow-contracting muscle cell populations, which differ by extensive sets of paralogous structural protein genes. We find that the regulatory gene set of the slow cnidarian muscles is remarkably similar to the bilaterian cardiac muscle, while the two fast muscles differ substantially from each other in terms of transcription factor profiles, though driving the same set of structural protein genes and having similar physiological characteristics. We show that anthozoan-specific paralogs of Paraxis/Twist/Hand-related bHLH transcription factors are involved in the formation of fast and slow muscles. Our data suggest that the subsequent recruitment of an entire effector gene set from the inner cell layer into the neural ectoderm contributes to the evolution of a novel muscle cell type. Thus, we conclude that extensive transcription factor gene duplications and co-option of effector modules act as an evolutionary mechanism underlying cell type diversification during metazoan evolution.

Motility is a hallmark of all animals, facilitated by the existence of contractile cells commonly known as muscle. In vertebrates, for instance, three distinct muscle types are present: striated skeletal, striated cardiac, and smooth muscles (Fig. 1a). Interestingly, despite their ultrastructural similarity, the two vertebrate striated muscles, skeletal and heart muscle, employ quite distinct regulatory gene sets, whereas smooth and cardiac muscles share several of the transcription factors, suggesting a common evolutionary origin[1–3]. Cnidarians (sea anemones, jellyfish, and corals), the sister group of bilaterians, also possess contractile cells of unclear relationship to their bilaterian counterparts. Thus, to reconstruct the evolutionary relationships and diversification of muscle cell types, it is necessary to unravel the transcriptional profile and the physiological features of muscle cells in cnidarians and to compare them to bilaterians (Fig. 1a, b).

All cnidarians are diploblastic, i.e., they develop from only two germ layers, commonly termed ectoderm and endo(meso)derm. In recent years, the sea anemone *Nematostella vectensis* has become a model among cnidarians[4]. In *Nematostella*, which belongs to the class

[1]Department of Neuroscience and Developmental Biology, Faculty of Life Sciences, University of Vienna, Djerassiplatz 1, 1030 Vienna, Austria. [2]Research platform Single Cell Regulation of Stem Cells, University of Vienna, Djerassiplatz 1, 1030 Vienna, Austria. [3]Institute of Molecular Biotechnology, Dr.-Bohr-Gasse 3, 1030 Vienna, Austria. [4]Michael Sars Centre, University of Bergen, Thormøhlensgate 55, 5008 Bergen, Norway. [5]Max Perutz labs, University of Vienna, Dr.-Bohr-Gasse 9, 1030 Vienna, Austria. [6]These authors contributed equally: Alison G. Cole, Stefan M. Jahnel. ✉e-mail: alison.cole@univie.ac.at; ulrich.technau@univie.ac.at

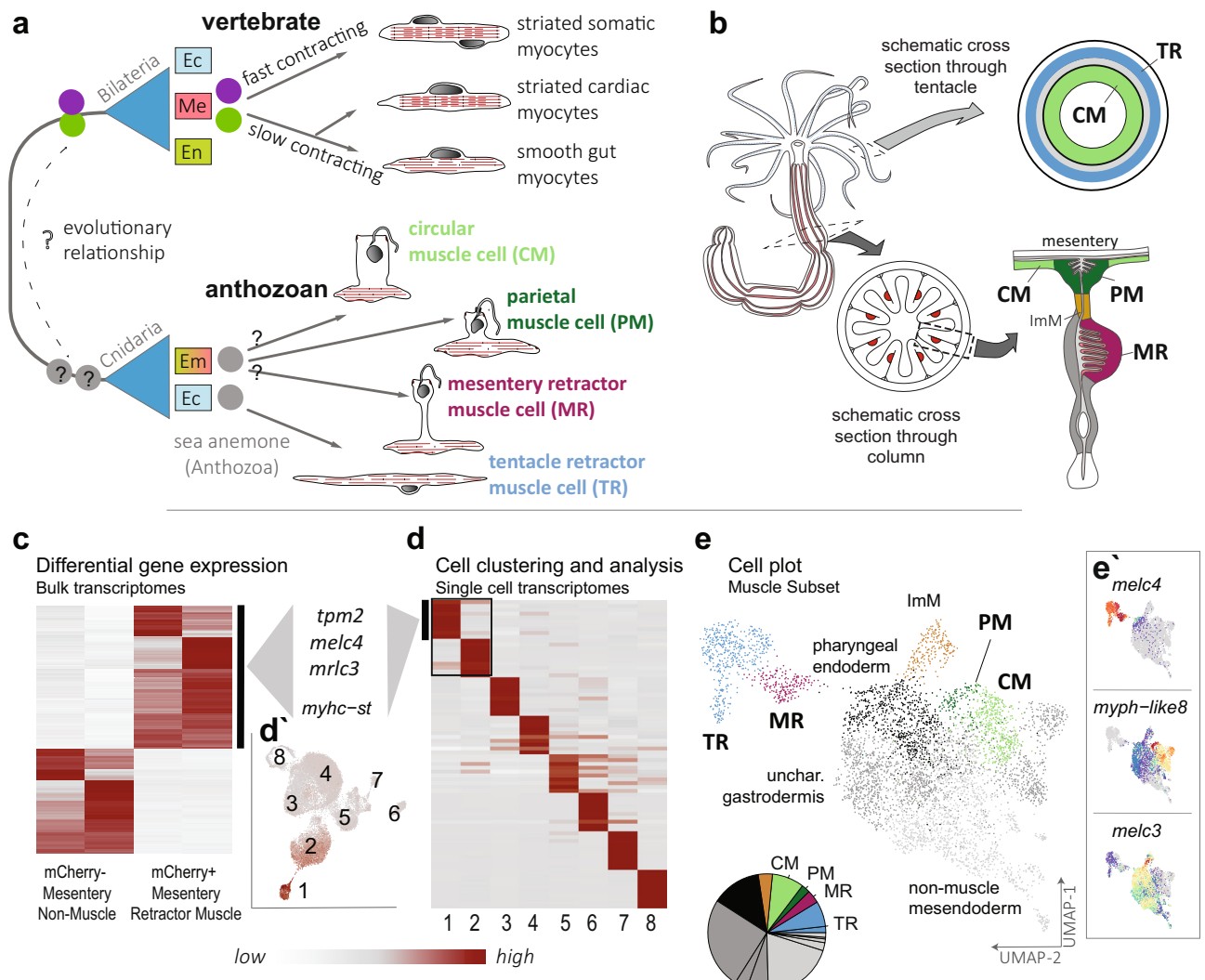

**Fig. 1 | Distinct muscle cell types in the sea anemone *Nematostella vectensis* are identified from single-cell sequencing data. a** Muscle cell relationships in vertebrates (Bilateria) and the sea anemone (Cnidaria). Two ancestral cell types corresponding to a fast (purple circles) and slow (green circles) contracting phenotype derive from the mesoderm in vertebrates. Muscles can arise from both cell layers in the diploblastic sea anemone, but their contractile properties are undescribed (gray). Both the intra- and interspecies evolutionary relationships of these muscle types are unclear. Ec: ectoderm, Me: mesoderm, En: endoderm, Em: endomesoderm **b** Schematic view of *Nematostella* muscle systems: tentacle and mesentery anatomy are illustrated schematically in cross section with the positions of the muscles indicated; blue: tentacle retractor (TR); light green: circular muscle (CM); dark green: parietal muscle (PM); red: mesentery retractor (MR); ocher: the intermuscular membrane (ImM) **c** Heatmap of differentially expressed genes of the mesentery-derived bulk transcriptomes. Average gene expression from differentially expressed genes with less than a twofold change of expression between library source (mCherry negative non-muscle, two replicates, vs. mCherry+ mesentery retractor muscle cells, two replicates) is imaged. The indicated muscle-related genes are upregulated in the muscle-cell libraries. **d** Heat map of differentially expressed genes across eight cell populations identified by single-cell RNAseq. The same muscle-related genes are detected in the retractor (1) cluster. Cell clusters carried forward as the muscle subset are indicated in the box. **d'** Dimensional reduction cell plot (UMAP) of the full dataset showing expression of the muscle marker myosin heavy chain (*myhc-st*) in the retractor (1) as well as the gastrodermis (2). **e** UMAP cell plot of the muscle subset annotated according to cluster identity. Four differentiated muscle cell clusters are identifiable and color-coded as in part (**b**). Differentiated muscle cells represent approximately one-quarter of the data subset (pie chart). **e'** Expression profiles of markers indicative of the retractor muscles (*melc4*), the bodywall muscles (*myph-like8*), and the intermuscular membrane (*melc3*), with rainbow expression profile: gray: no expression; blue: low; red: high.

---

of Anthozoa (i.e., corals and sea anemones), the gastric cavity is compartmentalized into eight inner epithelial folds, the mesenteries, which are derived from both germ layers[5]. Epitheliomuscular cells (epithelial cells with a basal contractile part[6]) can develop from both layers. F-actin staining and ultrastructural studies have previously described five morphologically distinguishable muscles: eight pairs of longitudinal endo(meso)dermal muscles: the parietal (PM) attached to the bodywall and retractor epitheliomuscular cells (MR) embedded within the mesenteries; the ring epitheliomuscles of the endo(meso) dermal lining of the body column and tentacles, referred here together as the circular muscle group (CM);[7] the longitudinally oriented

subepithelial tentacle retractor muscles (TR) that derive from the ectodermal epithelia of the tentacles (Fig. 1a, b). However, whether these muscles were the result of a cell type diversification and represent distinct muscle cell types or whether they are just different flavors of the same single muscle cell type is unclear.

Here we use single-cell transcriptomics to confirm the identity of four distinct muscle cell subtypes, which fall into two main categories, termed slow and fast-contracting muscles. Fast and slow-contracting muscles each share the expression of a set of distinct paralogous structural protein genes, which may convey the different contraction speeds. However, despite the shared effector gene set, the two fast-

contracting muscle cell types (MR and TR) are governed by different sets of transcription factors, including lineage-specific paralogous basic Helix-Loop-Helix (bHLH) transcription factors. This suggests that gene duplications of bHLH transcription factors and effector genes crucially contributed to the diversification of muscle cell types in these cnidarians.

## Results

### Transcriptomic profiling reveals four molecularly distinct muscle cell populations

In order to assess the molecular profile of cnidarian muscles in an unbiased way, we sought to generate a robust transcriptomic profile of muscle cells from the sea anemone *Nematostella vectensis*. We first generated bulk transcriptomic profiles from retractor muscle versus non-muscle cells of the mesentery upon dissociation and fluorescence-activated cell sorting (hereon called BULK dataset), using a retractor-muscle-specific transgenic line that expresses mCherry in the retractor muscles of the mesentery and tentacles under the control of a myosin heavy chain-striated type (myhc-st) promoter[8]. Confirming previous work[9], we found numerous muscle-related genes (e.g., myhc-st, myosin essential light chain (melc)s, myosin regulatory light chain (mrlc)s, tropomyosins (tpm)s, calmodulins, and calponins) enriched in the library of transgenic mesentery retractor (MR) sorted cells (Fig. 1c, Supplementary Data 1.1). To test whether these data are representative of all muscle types, we queried the adult tissue subset of our single-cell RNAseq atlas[10] (Supplementary Fig. 2). To distinguish between the parietal and circular muscle of the bodywall, we extended this dataset with the addition of an interparietal bodywall tissue sample. Analysis of the entire dataset revealed two clusters that are enriched for myHC-st expression (Fig. 1d′), as well as many other muscle-related genes, corroborating a recently reported single-cell dataset using a different scRNAseq platform[11] (Fig. 1d, Supplementary Data 1.2). We identified one cluster as retractor muscle according to overall similarity with the bulk transcriptome data from (FACS) sorted MR cells (Supplementary Fig. 2d). The other cluster was composed of cells harboring a gastrodermal signature, including the marker *snailA* (Supplementary Data 1.2[12]). Further analysis of this subset (retractor muscle + gastrodermal cell clusters) revealed molecular profiles corresponding to four distinct muscle cell populations (Fig. 1e′), while the majority of cells fall within populations with non-muscle-related gene profiles (Fig. 1e black through gray, Supplementary Fig. 3, Supplementary Data 1.3). Spatial expression profiles for genes enriched in muscle clusters were examined by in situ hybridization (Fig. 2b, Supplementary Fig. 4). Combined with further evaluation of the spatial distribution according to the tissue library of origin of cell clusters (Supplementary Fig. 3a, b) we were able to assign muscle cell clusters identities that correspond with anatomically recognizable muscles: tentacle retractor muscle (TR: blue), mesentery retractor muscle (MR: dark red), parietal muscle (PM: dark green), and circular muscle (CM: light green; Fig. 1e). One related non-muscle cell cluster is recognizable as the intermuscular membranous tissue between the RM and PM, by the expression of markers such as myosin essential light chain 3 (*melc*3 Fig. 1e′, Supplementary Fig. 4) and *gata*5 (Supplementary Fig. 3f).

### *Nematostella vectensis* muscle expresses two classes of effector modules

We next evaluated the transcriptomic profiles of each muscle population for their use of genes encoding for structural proteins. This gene set will include proteins involved in muscle function, representing the 'effector' gene set, or module, for each muscle cell type[13,14]. Notably, the ectodermal tentacle retractor muscle (TR) and the endodermal mesentery retractor muscle (MR) largely share the same set of effector genes (Fig. 2, Supplementary Figs. 2d, 5, Supplementary Data 1.4), while the parietal (PM) and circular muscle (CM) share a distinct set of genes, which are paralogous to the gene set from that of TR/MR (Fig. 2,

Supplementary Figs. 3b, c, 5). Differential expression of these two gene sets was validated by in situ hybridization, clearly uniting the tentacle and mesentery retractor muscles on one hand and the parietal and circular musculature of the bodywall on the other (Fig. 2b, Supplementary Fig. 4). This is remarkable, given the anatomical similarity between the longitudinally oriented mesentery retractor and parietal muscle cells, which both show a strong basal contractile and an apical epithelial part connected by a thin and flexible middle part[7]. Notably, the bodywall gene set is detected throughout most of the gastrodermis, albeit at lower levels (Supplementary Figs. 3e, 5a). Importantly, phylogenetic analyses of these structural protein paralogs revealed extensive gene duplications within the cnidarian and/or bilaterian lineages in all cases (Fig. 2e, Supplementary Fig. 6), independent of gene duplications that occurred within the Bilateria. We next asked whether the expression of these distinct paralogs of structural proteins correlates with the physiological properties of the muscle classes. Indeed, we found that the contraction speed of the TR/MR class is about 50× higher than that of CM muscles (Fig. 2d, Supplementary Fig. 7, Supplementary Data 2). In vertebrates, fast and slow twitch fibers within the somatic striated muscles are also characterized by specific expression of isoforms of Myhc[15,16], and so we presume that paralogs of structural proteins observed here convey specific properties, e.g., differential contractile force and speed. Given the strong overlap in their sets of structural proteins, we therefore refer to TR/MR and CM/PM as fast and slow muscles, respectively, even though we were unable to measure the contraction speed of PM alone due to its anatomical connection with CM. These data demonstrate that the four muscle cell populations in *Nematostella* comprise two discernable and physiologically distinct muscle classes that are characterized by specific combinatorial sets of paralogous, subfunctionalized structural protein-coding genes.

### *Nematostella vectensis* endodermal muscle regulatory signature is reminiscent of the bilaterian cardiac signature

A recent model for reconstructing cell type relationships hypothesizes that related cell types use the same core regulatory complex (CoRC), a collection of physically interacting transcription factors that together specify the terminal phenotype of a cell[13]. We next aimed to determine whether the transcription factor profile from either fast- or slow-muscle subtypes showed similarities to the regulatory profiles of the bilaterian muscle. We investigated the set of genes containing DNA-binding domains (regulatory signature) associated with each cell cluster using both candidate genes (reviewed in refs. [17,18]) and unbiased approaches (see "Methods" and Supplementary Fig. 8; Supplementary Data 1.4–7). While some candidate transcription (co-)factors commonly associated with muscle formation in Bilateria (e.g., *srf, mef2,* and *myocardin*) are detected in all four muscle cells types, they are also detected within non-muscle ectodermal derivatives at equivalent or even higher (e.g., *mef2*) levels (Fig. 3a: Muscle candidates), as has been previously documented[19]. Hence these genes have a much broader tissue distribution in *Nematostella* than in their bilaterian counterparts but may have a role in muscle development in conjunction with more specific factors.

The circular (CM) and parietal (PM) bodywall muscles are most similar to one another in their regulatory signature, commonly expressing *hand2, nkx2.2 A*, members of the Tbx20 family (*tbx20.1, tbx20.2*), but differing in their use of Tbx1/10 family genes (circular: *tbx1/10.1*; parietal *tbx1/10.2*) The longitudinally oriented parietal muscle is distinct from the perpendicularly oriented circular muscle by the expression of the homeodomain protein gene *VAX-EMX-like* and the SCL/TAL1 bHLH-family *tal1-like* (Fig. 3a, b: Bodywall Slow M, Supplementary Fig. 3f). Independent validation of these data have recently been generated, demonstrating that double knockdowns of the two Tbx20 paralogs that are expressed in both muscle types result in aberrant bodywall muscle formation[20]. We note that the shared

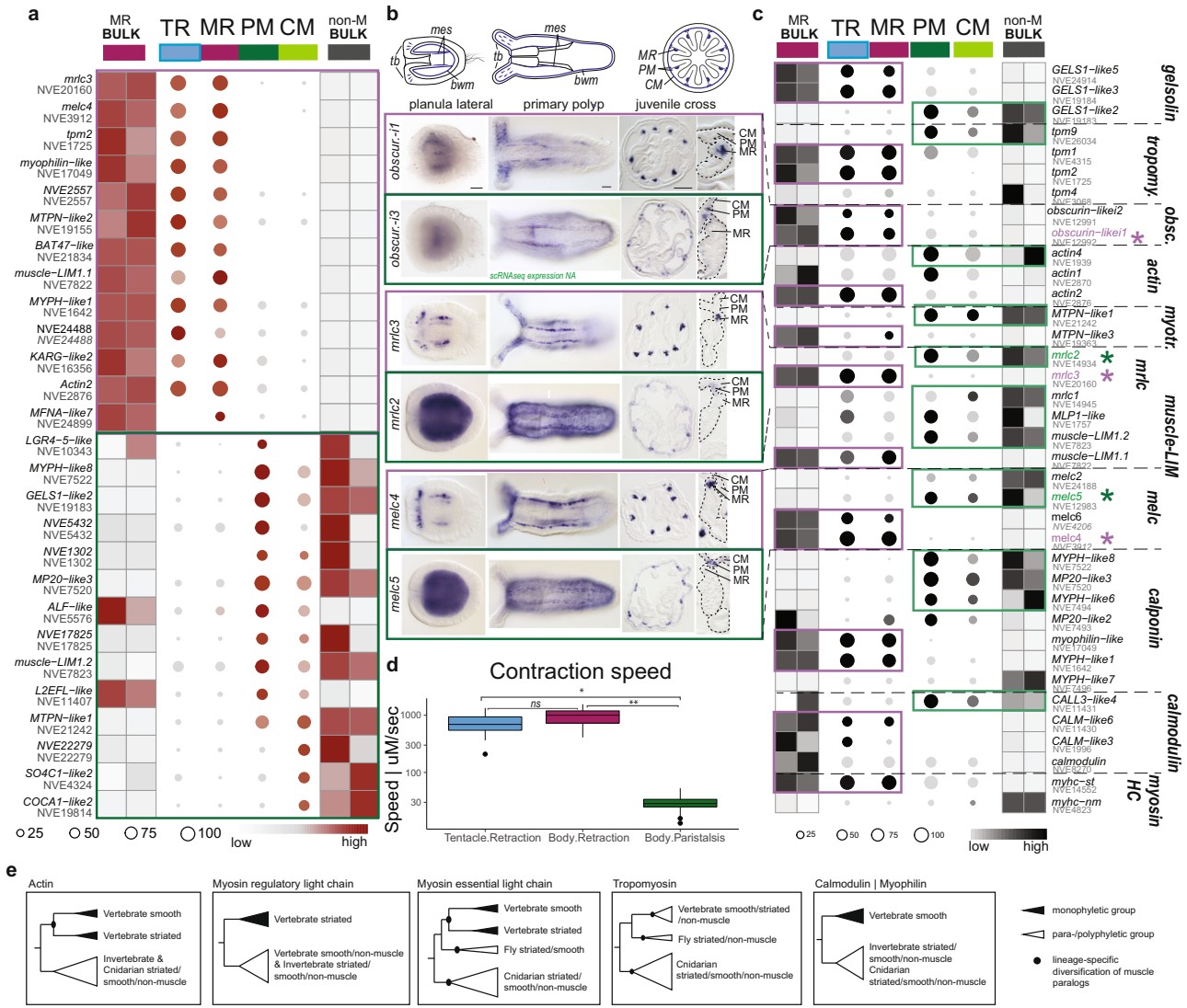

**Fig. 2 | Structural protein gene expression reveals two functional classes of muscle cells. a** Dot plot showing expression profiles for the 10 most significant differentially expressed genes for each muscle cluster. Cluster identity is indicated by the same coloration shown in (Fig. 1b, e). The relative expression profile of the same gene set within the bulk dataset of the non-muscle cells (non-M BULK) and the muscle cells (MR BULK) are illustrated as square blocks. Common gene sets that unite the retractor muscle (TR and MR) and the bodywall musculature (PR and CM) are boxed in purple and green, respectively. CM: circular muscle; MR: mesentery retractor; PM: parietal muscle; TR: tentacle retractor. **b** Validation of paralogous muscle gene sets by in situ hybridization. Schematic (top) and in situ hybridization for genes specific to the retractor muscles (purple boxes) or body-wall muscles (green boxes) are shown. One labeled mesentery is shown for each gene, and scRNA expression profiles are indicated in (**c**). *bwm*: bodywall muscle (PM & CM); *mes*: mesenteries; *tb*: tentacle buds. The oral pole is to the left in all whole mount images. Scale bars are 50 μm. **c** Expression profile of paralogous

genes illustrating differential use across clusters, set-up as in (**a**). Fast (purple boxes) and slow (green boxes) paralogs are highlighted. Genes shown in (**b**) are indicated with an asterisk (\*). **d** Measured contraction speeds for tentacle and body column retraction (blue and red) versus peristaltic bodywall contractions (green). Data are presented as a boxplot of log10 values of measured contraction speeds, illustrating the median, first, and third quartiles, with whiskers indicating the 95% percentile of the data. "\*" denotes *p*-value < 0.001 in paired two-sample Student's *t*-test (6e-27 and 3e-30), *ns* = nonsignificant (*p* = 0.078); *n* = 17 (body column retraction), 18 (tentacle retraction), and 51 (peristaltic contractions) independent measurement observations. See Supplementary Data 2−*t*-test for the full output of the statistical test. **e** Schematic overview of reconstructed gene family relationships is shown for select muscle proteins. Diversification of these protein families occurred independently in vertebrates and cnidarians. See Supplementary Fig. 6 for full-resolution trees.

combination of genes, together with the more globally expressed *srf/mef2*, is remarkably similar to the cardiomyocyte-defining set of transcription factors in bilaterians (GATA/Pannier/Serpent, Hand, NKx2.5/Tinman, Tbx5, Tbx1/10, Tbx20), in line with recent findings[5,21]. Further, while *gata* reads are detected in the parietal muscle, this gene is more strongly detected within the retractor muscles, as well as in non-muscle mesentery tissue and neuronal cells[5,22] (Fig. 3a). Of the slow-muscle regulatory profile, the *nk2/4* ortholog (*nkx2.2* A) and a *DMRT1* ortholog (*DMRT-F*) unite all endo(meso)dermal muscles (Fig. 3a), including the fast-contracting mesentery retractor muscle.

## *Nematostella vectensis* retractor muscle regulatory signatures suggest convergent evolution of fast-contracting myocytes

We next investigated whether the two fast retractor muscle populations (MR and TR) share a set of transcription factors. Surprisingly, we find that the two fast muscle cell populations are distinct with respect to their developmental regulators, sharing only the expression of *gata* and *e-protein*. Interestingly, the TR also expresses *elav*, a marker of a large subpopulation of neurons in *Nematostella*[11,23], as well as the neuronal transcription factors *soxB2a* (aka *soxB(2)*[24], *isl*, *otxA*, and *foxL2*, and a *metabotropic glutamate receptor* (Fig. 3a: Tentacle RM, 3b,

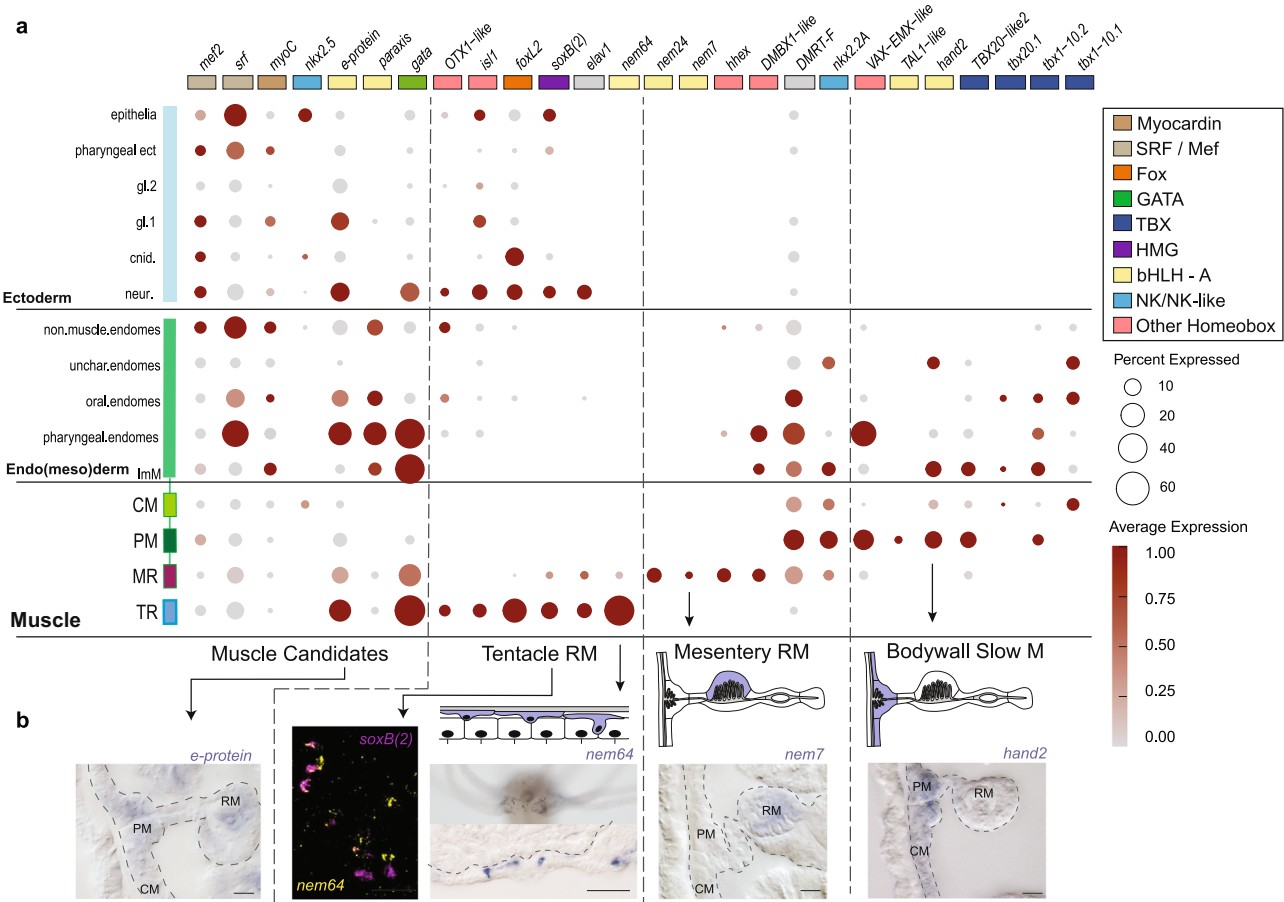

**Fig. 3 | Regulatory profiles of muscle cells indicate bHLH complex as key to cell type individuation. a** Dot plot showing relative expression profiles of selected regulatory genes across the entire dataset. Ectodermal derivatives are indicated in blue. Gene family relationships are indicated according to the color scheme shown in the legend. Gene sets are organized according to expression profile (Muscle Candidates, Tentacle RM, Mesentery RM, Bodywall Slow M). The relative positions of the muscle cells are shown schematically below the dot plot. **b** Validation of selected profiles by in situ hybridization. Scale bars are 20 μm.

Supplementary Fig. 8e, f). Thus, on the level of the transcription factors, the tentacle retractor cells are more reminiscent of neurons than of muscle, which may reflect their common ectodermal origin from *soxB(2)*-positive progenitor cells[10]. This correlates with their anatomy and development: unlike the epitheliomuscular nature of the endodermal muscles, the ectodermal tentacle retractor muscles form by apical detachment from the epithelium resulting in basi-epithelial positioning, similar to ganglion neurons[7]. Thus, despite their use of a common effector gene set, the two fast muscles employ distinct sets of developmental regulators and have distinct ontogenetic origins in different germ layers. This demonstrates that with respect to transcription factors, the fast mes(endo)dermally derived mesentery retractor muscle (MR) shows greater similarity to the slow muscle (PM, CM) than to its ectoderm-derived counterpart in the tentacle (TR).

## Cnidarian muscle specification is linked to bHLH gene expression

In the development of vertebrate striated muscles, the paralogous myogenic regulatory factors (MyoD, Myf5, Myf6, Myogenin) are crucial muscle determinants[25]. Yet, there are no orthologs of this family within cnidarians. However, we find that the bHLH factor coding genes *nem7*, *−24*, and *−64*, which form a sea anemone-specific paralog group with no clear orthology relationship to any of the bilaterian bHLH families[26], are specifically expressed in the muscles in *Nematostella*. These three genes (*nem7*, *nem24*, and *nem64*) are clustered on chromosome 2 of the Vs.2 *Nematostella* genome[27], together with *paraxis* and *twist* (Fig. Supplementary Fig. 9), suggesting that these genes might have evolved

by tandem gene duplications. Indeed, when we assessed their phylogenetic relationships, we found that the sea anemone bHLH paralogs branch within a superclade of the a-bHLH family together with Paraxis and Twist, and Hand—three well-characterized proteins involved in muscle differentiation in bilaterians. Hence we refer to this gene clade as the Cni-PaTH genes (Fig. Supplementary Fig. 9)[26]. Notably, we observe a subfunctionalization between the paralogous genes: the mesentery retractor muscles specifically express *nem24* and *nem7* (Fig. 3a, b: Mesentery RM; Supplementary Fig. 8), while the ectodermally derived tentacle retractor muscles specifically express *nem64*. Tissue-specific bHLH proteins have been described to form heterodimers with more ubiquitously expressed bHLH proteins of the E-protein group. We detected a homolog of the e-protein gene, which is expressed in all four muscles, suggesting that it might be acting as a partner to the specific bHLH factors. The shared expression of the bHLH cofactor gene e-protein and the presence of individual bHLH paralogs is remarkably reminiscent of the role of bHLH proteins in muscle formation in vertebrates, yet, with nonhomologous genes. To test the function of the bHLH transcription factors, we next generated CRISPR-Cas9-mediated F2 homozygous mutants of global muscle bHLH cofactor gene *e-protein* and of the tentacle retractor muscle-specific gene *nem64* (Supplementary Fig. 10). Knockout of *e-protein* severely disrupts mesentery patterning and eliminates fast muscle markers at the planula stage (Fig. 4a). This indicates that e-protein is required for the formation of all muscles, supporting the idea that it acts as a common partner of more specific bHLH TFs. Conversely, knockout of the tentacle-specific bHLH TF gene *nem64* selectively

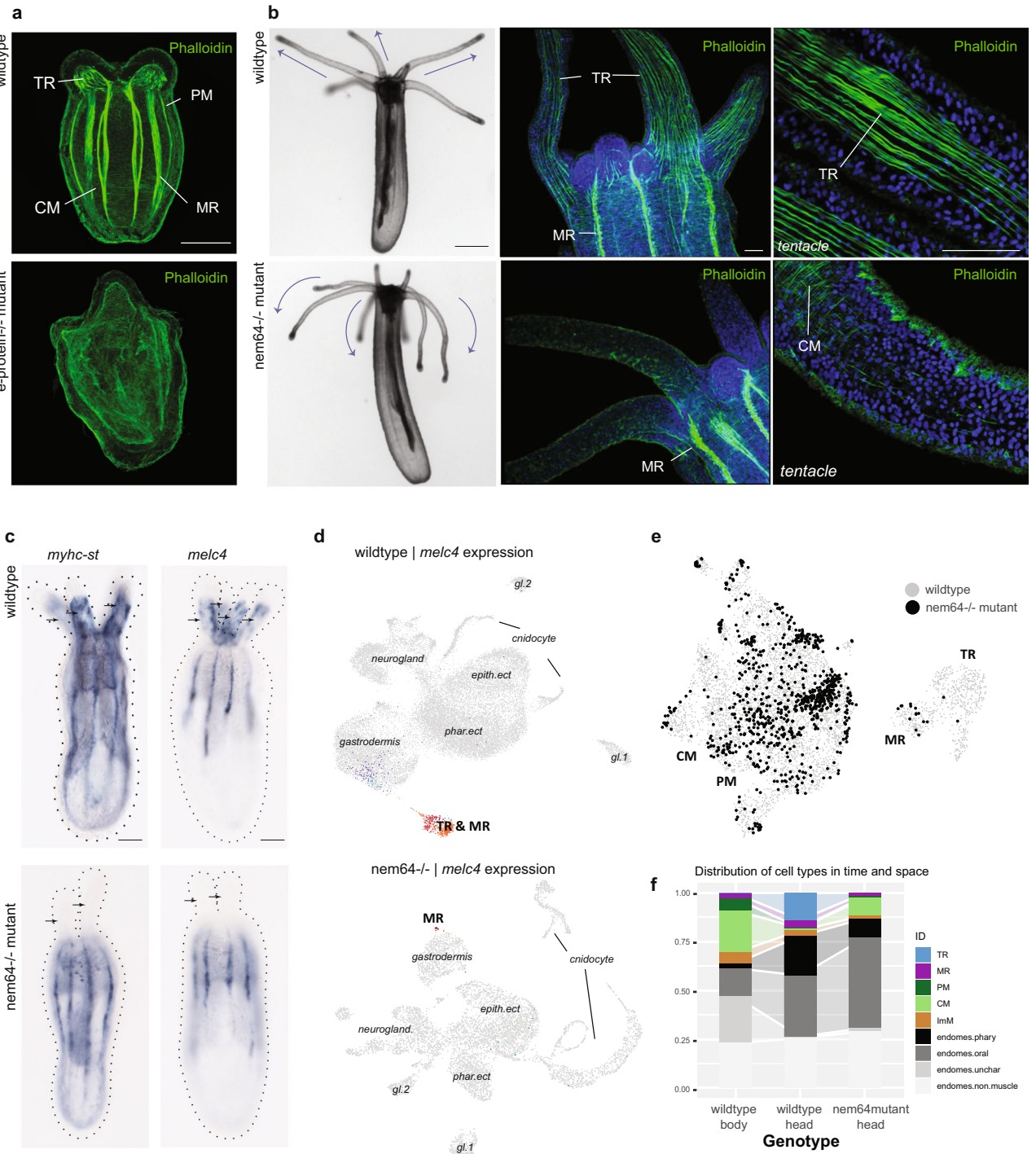

**Fig. 4 | Gene perturbation selectively eliminates muscle.** Wild-type animals are shown in the upper row panels and mutant animals in the lower row panels. **a** *e-protein*−/− mutant: Phalloidin staining in control (upper) and mutant (lower). **b** *nem64*−/− mutant: Control animals (upper) show extended tentacles with intact phalloidin-positive retractor muscles, whereas mutant animals (lower) exhibit droopy nonresponsive tentacles and the absence of phalloidin-positive muscle within the tentacle retractors. **c** Structural marker genes (*myhc-st, melc4*) are lost within the tentacles of *nem64*−/− mutant animals (bottom). Outline of animal indicated with a dashed line, arrows: tentacles. **d** Single-cell sequencing of *nem64*−/− animal heads lose the retractor muscle cluster present in the wild type (TR & MR: top), but the retractor muscle signal is present in a few cells of the gastrodermis (MR). **e** Mapping mutant cells onto the wild-type UMAP confirms the presence of MR cells but the loss of TR cells. **f** The distribution of cell types is otherwise similar between wild type and mutant heads. Scale bars are 100 μm (embryos) and 20 μm (confocal images).

eliminates the ectodermal retractor muscles of the tentacles, whereas the bodywall muscles and mesentery retractor muscles are unaffected. *Nem64*−/− mutants form polyps with a normal morphology but are unable to move their tentacles (Fig. 4b). This mutant also demonstrates that the longitudinal ectodermal muscle is not necessary for tentacle formation, in contrast to bodywall elongation, which has been shown to require proper longitudinal muscle formation[20]. We further validated the absence of the tentacle retractor muscle via in situ hybridization of genes of the fast-muscle identity (*myhc-st, melc4*), demonstrating that while the mesentery retractor remains intact, these

animals have selectively lost the tentacle-derived ectodermal counterpart (Fig. 4c). To further confirm the absence of the tentacle muscle signature we prepared single-cell RNA libraries from *nem64−/−* heads cut below the pharynx, where in wild-type animals both the oral mesentery retractor and the tentacle retractor muscles are present. In this way, we enriched for cells of the tentacles while also retaining the ability to capture the mesentery retractor attached to the pharynx. Within the mutant library itself, we found no distinct fast retractor cell cluster but detected the fast-muscle markers in a small subset of cells within the endo(meso)dermal cluster that corresponds to the mesentery retractor muscle cells (Fig. 4d). We mapped the endo(meso)dermal cells of the *nem64−/−* head tissue onto the reference wild-type endo(meso)dermal subset, and confirmed that the head tissue of *nem64−/−* animals retain the fast-contracting muscle signature with MR-identity (e.g., *nem24/dmbx* regulatory signature as well as the MR-specific *NVE8057*), while lacking specifically the TR cluster and corresponding marker genes (e.g calmodulin *CALM-like3*). (Fig. 4e; Supplementary Fig. 10c). The distribution of other cell types remains similar between the wild type and mutant (Fig. 4f). These data indicate that expression of *nem64* within individual cells of the tentacle ectoderm is necessary for specification of the TR subset of muscles. Altogether these data suggest that the specification of distinct muscles in the sea anemone is mainly driven by bHLH transcription factors using a mixture of conserved (*e-protein, paraxis, hand*) and independently evolved factors (*nem7, −24, −64*).

## Discussion

Here we show that the non-bilaterian, diploblastic sea anemone has diversified its contractile cells into four muscle cell types with distinct transcriptional profiles (Fig. 5a). Our data provide evidence for the presence of two distinct structural profiles, or effector modules, which correlate with 'fast' and 'slow' contractile properties. While the presence of a bilaterian cardiac regulatory gene network has been previously noted[5,28], we demonstrate here that these genes are used not only in germ layer specification but also in the determination of specific muscle subtypes. Thus sea anemone slow muscles are likely to share a common ancestry with the bilaterian cardiac/smooth muscle cell type. By contrast, *Nematostella* fast-contracting muscles show little similarity with the striated muscles of bilaterians nor with the slow-contracting muscle cells in *Nematostella*. Thus, in line with our previous study[9], we propose that fast-contracting muscles, being smooth or striated, are likely to have evolved independently in cnidarians and bilaterians.

Our data suggest that the diversification of muscle cell types found in this cnidarian was accompanied and perhaps facilitated by extensive gene duplications and subfunctionalization. The fast and slow muscles are characterized by >25 specific paralogs from at least nine gene families coding for several effector genes encoding structural proteins. Interestingly, in *Nematostella*, the two fast muscles (tentacle and mesentery retractor muscle) employ distinct sets of developmental regulators while expressing largely the same structural protein-coding genes (Fig. 5a). Thus, the level of regulatory genes is

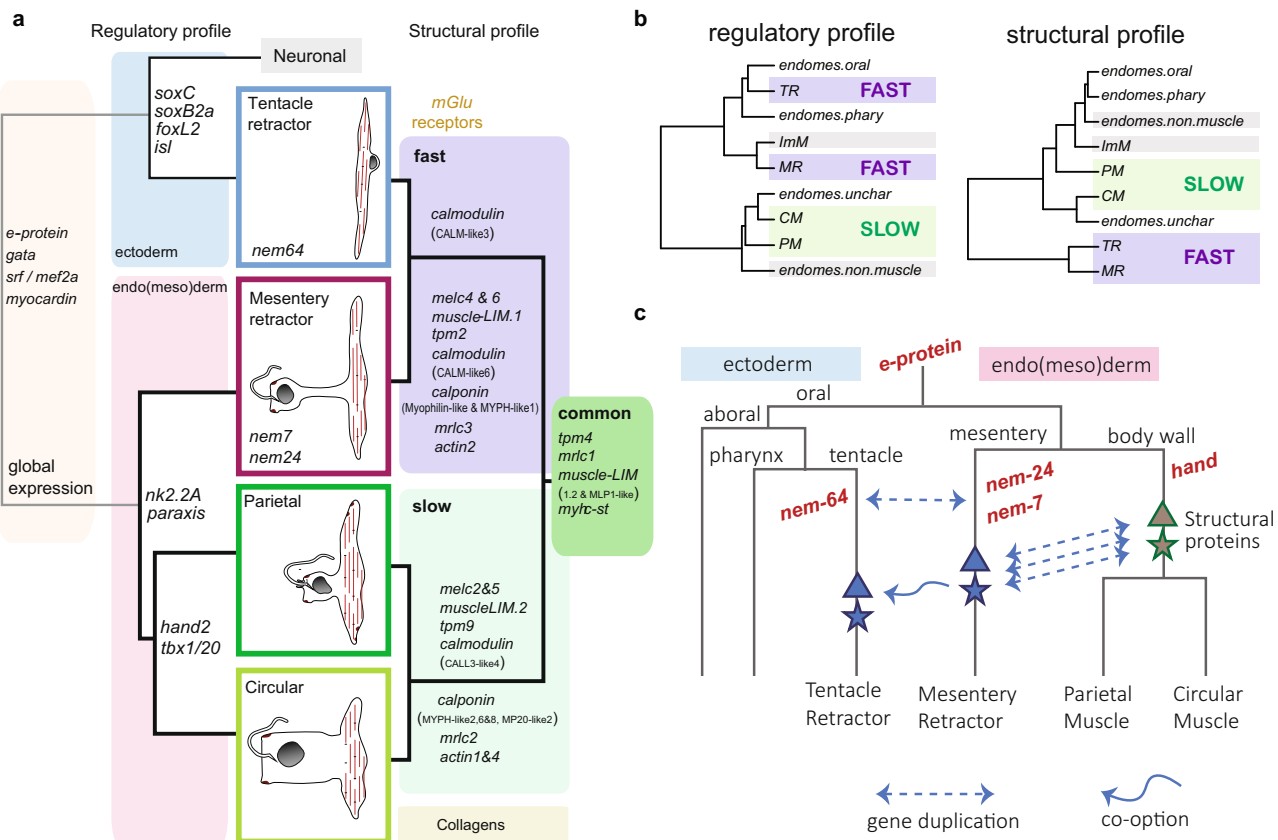

**Fig. 5 | Summary of anthozoan muscle transcriptome and model of muscle cell evolution. a** The left side shows key regulatory genes and their association with the four muscle types shown in the center panel. The dendrogram on the left indicates the developmental relationships between the cell types. Individual genes within the cell type boxed in the center panel are proposed here as cell-type-specific regulatory factors, whereas those positioned on the dendrogram are shared between descendants of these branches. On the right side, the described effector modules, or aponemes, are indicated. Note that the composition of this list is identical, but the individual paralogs are specific to each effector module.

**b** Hierarchical clustering with expressed DNA-binding proteins vs. structural proteins shows noncongruence between regulatory gene profile and effector gene profile among the four muscle cell types in *Nematostella*. **c** Hypothesized events underlying the evolution of myocytes in the sea anemone. The fast-contracting effector module is hypothesized to have evolved within the endo(meso)dermal muscle populations and was then co-opted by the ectodermal epithelium after the radiation of PaTH-related bHLH proteins (**Pa**raxis, **T**wist and **H**and (see Supplementary Fig S1.5), red) and recruitment of one bHLH TF paralog, nem64, to the tentacle ectoderm within the sea anemone lineage.

not congruent with the effector genes (Fig. 5b). In this context, it is striking that the two retractor muscles specifically express lineage-specific paralogs of bHLH proteins, i.e., nem64 in the tentacle retractor muscle and nem7/nem24 in the mesentery retractor muscle. It is feasible that specialization for fast retraction occurred first in the ectoderm of an ancestral cnidarian lineage, where even striation is present[9,29]. Indeed, the expansion of the PaTH family is specific to the anthozoans, as medusozoan cnidarians have a single ortholog (Supplementary Fig. 9). In this scenario, the TR examined here is simply the remnant of a once more wide-spread ectodermal muscle. However, the tentacle retractor muscle is unique not only in that it is the only ectodermal muscle, but it also displays a number of morphological and molecular features that are similar to neurons. In an alternative scenario, nem64 came to be expressed in the ectodermal epithelial cells of the tentacle only after the expansion of this gene family. Once there, nem64 recruited the retractor muscle-specific bHLH-binding sites of the "fast" structural protein genes, which in the mesentery retractor muscle are bound and regulated by its paralogs nem7 and nem24 (Fig. 5c). In this case, the TR represents a novel cell type arising as a hybrid between neurons and muscles. The use of different regulatory gene sets to activate similar effector gene sets has been well documented in flies[30] and worms (reviewed in ref. [31]). bHLH transcription factors may be particularly suited for this, as they act as dimers with e-proteins, yet are rather promiscuous in their binding properties[32,33]. Thus, in more general terms, paralogous transcription factors may easily co-opt even complex sets of target genes into a new cellular context facilitating the diversification of cell types.

## Methods

### Bulk transcriptome data processing and analysis

In order to enrich muscle tissue in *N. vectensis*, we used the muscle-specific myhc-st::mCherry reporter line[8], which specifically expresses mCherry in the retractor and tentacle longitudinal muscle of the polyp. Adult polyps (>4 cm length in a relaxed state) with strong mCherry expression in the retractor muscle region were chosen and relaxed for 30 min by adding a few drops of 7% $MgCl_2$ to a large Petri dish. Subsequently, animals were pinned down with needles, and the bodywall was opened from the pharynx to the foot. After lateral fixation of the opened bodywall with additional pins, as much mesenterial tissue as possible was removed distal from the retractor muscle. Tissue enriched for mCh⁺ retractor muscle was then removed by cutting through the intermuscular region and pooled, resulting in two replicates with muscle tissue derived from eight animals in each case. Tissue was dissociated similarly as described previously[34] by adding dissociation mix (50 µl papain (Sigma-P4762; 3.75 mg/ml in *Nematostella* medium (1/3 artificial seawater), 50 µl Collagenase (Sigma-C9407; 1000 U/ml in *Nematostella* medium), 3.5 µl 0.1 M DTT) to at most four retractor muscles at once in a 1.5-ml Eppendorf tube and left at room temperature overnight (12–15 h). The next morning tubes were flicked gently, and successful dissociation into single cells was examined under the microscope. Afterward, cells were spun down (300 G, 15 min), and the pellet was washed with *Nematostella* medium and resuspended carefully. This process was repeated twice and dissociated and washed cells were combined and stored at 4 °C until further processing (<1 h). Cells were sorted using FACSAriaII and collected in a 15-ml falcon tube containing TRIzol LS Reagent (Thermo Fisher Scientific), maintaining a final dilution of 3:1. In order to prevent degradation, sorted cells were mixed with TRIzol regularly. Samples were kept on ice, and total RNA was extracted according to the manual. Only samples with an RQI >7.3 were accepted for library preparation (NEB polyA) and subsequently sequenced (HighSeq 2500) single-end with a read length of 50.

Raw reads were processed using Cutadapt[35] for adapter trimming and SortMeRNA[36] in order to remove ribosomal RNA. Processed reads of *N. vectensis* were subsequently mapped to the repeat-masked genome using TopHat[37], and reads with a mapping quality <20 were discarded. Aligned reads were counted with HTSeq[38], and differential gene expression analysis was performed using edgeR[39]. Workflow is depicted in Supplementary Fig. 1f.

### Single-cell RNA sequencing

Five-month-old polyps were dissected into four separate tissues: tentacles, mesenteries, pharynx, bodywall (see Supplementary Fig. 1a:e). In a separate experiment, the tentacles were removed, and the entire pharyngeal mass was harvested, including the surrounding bodywall; in a third experiment, the bodywall was splayed open, and bodywall tissue in between the parietal ridges was harvested. Tissue pieces were treated with undiluted TrypLE™ Select (Thermo Fisher, A1217701). For 1 h, the tissues were allowed to disintegrate in the enzyme with gentle agitation on a shaker, then dissociation into single-cell suspensions was completed with occasional pipetting over the course of another 1.5 h. Cell viability and counts were assayed with a Cellometer X2 (Nexcelom), and suspensions were diluted to 1000–1700 cells/µl. Cell suspensions were kept on ice for no more than 1 h and loaded into a 10x Genomics single-cell platform using Vs.2 reagents. Libraries were generated following the manufacturer's protocol. Sequenced libraries were processed through the CellRanger 3.1.0 pipeline using default parameters and forcing the pipeline to recover 7000 cells for each library. Reads were mapped to a customized transcriptome[40], wherein all gene models were extended by 1000 bp in the 3′ direction or until the start of the next gene model in the same orientation.

### Single-cell transcriptomic analysis

The resulting count matrices were imported in R for further processing using the R-package Seurat Vs3 ([41,42]). The seven libraries were merged using the merge function, and the resulting dataset was then filtered for cells containing at least 200 genes, and outliers with UMI counts >5000 were removed, as these likely reflect cell multiplets (Supplementary Fig. 1g). Data were scaled to 5000 reads, log normalized, and 2000 variable genes were identified using the FindVariableGenes function. To facilitate inter-library comparisons, gene expression values were first standardized within each library, and these relative expression values were used for dimensional reduction and cell clustering (Supplementary Fig. 2a, b). Reduction algorithms were applied to the dataset (principal components analysis, UMAPs[43]), and hierarchical clustering was performed using all principal components with a standard deviation of >2. Cluster-specific genes sets were determined for each cell population using the Seurat FindAllMarkers function using all variable genes, requiring genes to be expressed in at least 10% of the cell population and showing a log fold change in expression of at least 0.6 and a *p*-value of less than 0.0001. The resultant gene lists were examined in order to assign a population identity to each cluster (Supplementary Fig. 2b, Supplementary Data 1.4). Muscle clusters were identified based on the presence of some pan-muscle markers, including but not limited to *myhc-st*, and this muscle-enriched subset of the full dataset was processed in a similar manner.

For the muscle-enriched subset, increasing cluster resolutions were evaluated, and the resolution was selected where the bodywall muscle separates into parietal and circular muscles. Cell cluster identity was assigned semi-automatically[10] after inspection of the Seurat::FindMarkers output. Briefly, the mean scaled expression from all cells of the cluster was calculated for a list of selected marker genes, and the highest calculated value for the cluster was used to assign a cluster ID. Clusters corresponding to cell states other than those identified as differentiated muscles were re-collapsed for the purposes of this study. Gene sets used to annotate the clusters are found in Supplementary Data 1.8. GO term enrichment was performed across all clusters (Supplementary Fig. 3d), and assessment of gene usage across clusters is visualized in terms of the number of cells detected and average expression levels using the DotPlot function (Supplementary

Fig. 3e). Specific structural genes were identified as having at least 20 reads within the cluster, and absent from the ectodermal portion of the dataset (<200 reads) (Supplementary Fig. 5). In the case of transcription factors whose overall detection was low, >5 reads associated with genes of interest within the cell population is taken as evidence of expression within that population as a whole. Muscle-specific genes were selected as being absent from non-muscle cell populations within the full dataset (<75 reads) (Supplementary Fig. 8). The R script for generating the data objects, analysis, and figures is provided on our GitHub page [https://github.com/technau/NemMuscle].

## CRISPR mutant lines

The template for *nem64* single guide RNAs was amplified by PCR based on the annealing of two oligos:[44] a T7- and guide RNA-encoding oligo (desalted): 5′-GAAATTAATACGACTCACTATAGTTGCTAACTCCGGTG AGACGGTTTTAGAGCTAGAAATAGCAAG-3′; invariant reverse primer (desalted): 5′-AAAAGCACCGACTCGGTGCCACTTTTTCAAGTTGATAA CGGACTAGCCTTATTTTAACTTGCTATTTCTAGCTCTAAAAC-3′. Guide RNAs were in vitro transcribed using a T7 MegaScript transcription kit (ThermoFisher), followed by ammonium chloride precipitation and diluted in nuclease-free $H_2O$. CRISPR Cas9-mediated mutagenesis was performed as described previously using 1.5 µg/µl nls-Cas9 protein (PacBio), 150 ng/µl guide RNA, and a modified buffer containing 220 mM KCl[45,46]. DNA was prepared from F1 generation individuals, and a 211-bp fragment containing the recognition sequence of the guide RNA was amplified using the following oligos: Forward-1 primer 5′-GAGCCAGTACGGCACACAAACC-3′ (also used for sequencing) and Reverse-1 primer 5′-TCGTGCGTTCGCTAATCTCCTC-3′. A 3-amino acid TAA insertion introducing a stop codon, and creating a new HpaI restriction site, has been detected. Further larger-scale genotyping was performed by HpaI restriction digest testing of a 319-bp fragment using Forward-1 oligo and Reverse-2 oligo 5′-CATCCAGATGA TCTCGGTTTTCG-3′.

CRISPR Cas9-mediated mutagenesis for *e-protein* was performed using the following oligos for single guide RNA generation: 5′ TAG-GACGGCATAACAGCTAGGGAG 3′, 5′ AAACCTCCCTAGCTGTTATGCC GT 3′. DNA was prepared from individual polyps, and a 574-bp fragment containing the recognition sequence of the guide RNA was amplified using the following oligos: 5′ CGAGATGGCCTGGACCAAAT 3′, 5′ AGCTTCACGTCACGTCTCTG 3′. Sequencing of the resultant fragments confirmed a 4-bp deletion that would result in a truncated messenger RNA lacking the bHLH-DNA binding domain (Supplementary Fig. 10 a, b).

## Phylogenetic analysis

Sequences used for phylogenetic analyses were either downloaded from UniProt (http://www.uniprot.org) or NCBI (https://www.ncbi. nlm.nih.gov) databases. Alignment was done via MAFFT[47], and maximum likelihood trees were calculated using IQ-TREE[48]. Neighbor-joining trees were generated with ClustalX[49]. Phylogenetic trees were subsequently visualized with FigTree v1.4.3. and processed with Adobe Illustrator. Sequence alignments are provided as a fasta file (Supplementary Data 3)

## Calculation of muscle retraction speeds

Juvenile polyps (4–8 tentacles) of the myhc-st::mCherry transgenic line[8] were used for all experiments. Animals were imaged with Nikon Eclipse TS100 equipped with a Nikon DS-Qi camera. The exposure and gain were set as short as possible to ensure the highest possible frame rate and sharp images. The animal was placed within the frame, and videos were recorded before and after a dilution of acetic acid 1:1000 in ELIX was used to trigger tentacle contraction, whereas a ratio of 1:100 was used to trigger a full body contraction. Videos were then processed with the imaging software ImageJ. The distance between the beginning and the end of the fluorescent (mCherry) mesentery

retractor was measured frame by frame, and the time point showing the greatest distance was used to calculate the contraction speed of the body column as a proxy for mesentery contraction rate. Similar measurements were made from the base to the tip of the tentacles in order to estimate tentacle retractor speed. To calculate peristaltic contraction, images were stabilized using the TrakEM2 plugin for Fiji; landmarks were used to measure the maximum and minimum circumference of the animals. Contraction speed was then calculated by multiplying the measured values by Pi. See Supplementary Fig. 7 for illustration.

## Fixation, whole-mount in situ hybridization, and imaging

Animals used for in situ hybridization were treated as described previously[5]. Briefly, polyps were relaxed prior to fixation by adding 1 M $MgCl_2$ to the culture media. Specimens were fixed for 1–2 min in 2.5% glutaraldehyde/3.7% formaldehyde in 16 ppt artificial seawater (*Nematostella* media: NM), followed by a 1-h fixation in 3.7% formaldehyde/NM at 4 °C. Specimens were washed in PBS and dehydrated and stored in MeOH at −20 °C for at least 24 h. Prior to hybridization, specimens were rehydrated to PBS and digested with Proteinase K (20 min in 10 µg ml⁻¹ for all larval stages, 20 min in 20 µg ml⁻¹ for juveniles), and post-fixed for 1 h in 3.7% formaldehyde. Embryos were blocked for 2 h in formamide hybridization buffer (50% formamide, 5× SSC pH 4.5, 1% SDS, 0.1% Tween 20, 100 µg ml⁻¹ heparin, and 5 mg ml⁻¹ Torula yeast RNA), or urea hybridization buffer for juveniles (4 M urea, 5× SSC pH 4.5, 1% SDS, 0.1% Tween 20, 100 µg ml⁻¹ heparin, and 5 mg ml⁻¹ Torula yeast RNA), without probe and then hybridized with 0.5 ng/µl or 2 µg/µl (for fluorescence) (see Supplementary Data 1.10 for primer sequences) in the hybridization buffer for 3 days at 63 °C or 60 °C (juveniles). Hybridization buffer for juveniles contained 5% dextrane sulfate and 3% blocking reagents (Roche Post-hybridization washes (all without dextrane sulfate of blocking reagent) in decreasing concentrations of SSC (0.1x SSC for fluorescence) were carried out at 63 °C or 60 °C until arriving into PBS/0.1% Tween20 at room temperature. For fluorescent detection, animals were washed with 0.1 M Tris-HCl pH 7.5/0.15 M NaCl/0.1% Tween 20 (TNT) rather than PBS/Tween20. Antibody concentrations used were: α-Dig-AP (1:2000 Roche 11093274910) for colorimetric, or α-Dig-POD (1:100; Roche 11633716001), and α-Fluo-POD (1:50; Roche 11426346910) for fluorescence. Alkaline phosphatase staining buffer for colorimetric detection was NaCl [100 mM], $MgCl_2$ [50 mM], Tris-HCl [100 mM, pH = 9.5], Tween20 [0.1%] with BCIP [1.5 µl/ml] + NBT [1 µl/ml]. Tyramide staining buffer for fluorescent detection consisted of 0.1 M boric acid pH 8.5/0.2% Triton X-100/20 µg ml⁻¹ 4-iodophenylboronic acid/2% dextrane sulfate (MW > 500,000) and 10 µg ml⁻¹ tyramide−DyLight549 or tyramide−DyLight488. Colorimetric detection was monitored under a stereo microscope until desired intensity was reached, then tissues were washed in EtOH and brought to glycerol for imaging. Fluorescent detection was initiated with 0.003% $H_2O_2$, left for 30–45 min, washed with TNT, mounted in Vectashield, and imaged on a Leica SP5 confocal microscope. Colorimetrically stained juveniles were treated as follows: samples were infiltrated with 10% gelatine in PBS at 37 °C for 30 min. After that, samples were transferred to a mold filled up with liquid gelatine solution and subsequently orientated longitudinally. Solidified blocks were then post-fixed in 3.7% formaldehyde at 4 °C overnight, washed in PBS, and sectioned at 20–30 µm using a Vibratome Leica VT 1200 S. Sections were mounted on slides with 86% glycerol, visualized and imaged on a Nikon 80i upright microscope. Images were processed (cropping, level adjustment) using Adobe Photoshop CC15. All figures were assembled, and schematics drawn in Adobe Illustrator CC15.

## Phalloidin staining

Animals were relaxed in 7% MgCl2 in *Nematostella* medium for 30 min and subsequently fixed in 3.7% formaldehyde in PBT (PBS, 0.4% Tween)

at 4 °C overnight and washed thoroughly the next day. Samples were then incubated in Phalloidin-AlexaFluor 488 (Thermo # A12379) (3 μl/ 100 μl PBT) and DAPI (1:1000 in PBT) overnight at 4 °C in the dark, washed and mounted on a glass slide in VECTASHIELD Antifade Mounting Medium and imaged on a Leica SP5 confocal microscope.

## Statistics and reproducibility

No statistical method was used to predetermine sample size; single cell data were excluded from the analyses as described above; cell capture via microfluidics is random; the experiments were not otherwise randomized; the investigators were not blinded to allocation during experiments and outcome assessment. All in situ hybridization experiments were repeated a minimum of three times with more than 50 embryos/3 juveniles each; images shown throughout this work are representative of the staining pattern for the probe.

## Reporting summary

Further information on research design is available in the Nature Portfolio Reporting Summary linked to this article.

## Data availability

Raw sequence data generated in this study have been deposited in the GEO database under accession code "GSE154477". The processed single-cell data are available via the "UCSC Cell Browser [sea-anemone-atlas.cells.ucsc.edu/]". The muscle contraction speed data generated in this study are provided in Supplementary Data 2. All other data are available in the main text and its Supplementary information files or from the corresponding author upon reasonable request. Source data are provided with this paper.

## Code availability

R script for generating the analysis presented in this manuscript is found on our GitHub page https://github.com/technau/NemMuscle [https://doi.org/10.5281/zendo.7509380].

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

## Acknowledgements

Michael Sixt, Alexander Leitner, and Doreen Milius provided access and technical support with the FACSAriaII machine. We are grateful to Hanna Kraus for the polyp schematic in Fig. 1b. Maximilian Haeussler, Matthew Speir, and Brittney Wick processed the data for storage and viewing on the UCSC Cell Browser. We thank Sasha Mendjan and Frank Schnorrer for critically reading a previous version of the manuscript. This work was funded by grants from the Austrian Science Fund FWF P27353 (U.T.), T814 (S.K.), and P31018 (A.G.C.).

## Author contributions

U.T. conceived the study and contributed to the analyses of all data. S.J. and B.Z. generated and analyzed the bulk transcriptome datasets. J.H., A.D., P.M., A.G.C., and J.S. generated single-cell transcriptome datasets. A.G.C. analyzed all transcriptome data. E.H., S.J., and P.S. performed in situ hybridization. S.J. and P.S. generated the mutant lines. R.R. performed the speed measurements. E.T., S.K., and P.S. generated the phalloidin images. U.T., S.K., and A.G.C. wrote the manuscript with input from all authors.

## Competing interests

The authors declare no competing interests.
