## [Peer Review File · Nature Communications]

REVIEWER COMMENTS

Reviewer #1 (Remarks to the Author):

In this paper, Cole et al aim to use single-cell sequencing to discern the muscle cell types in the sea anemone *Nematostella*, address the germ layer origin of these cell types, and compare with the bilaterian muscle cell types. They first identify four transcriptomically separable cell types, circular muscle (CM), parietal muscle (PM), tentacle retractor (TR) and mesentery retractor (MR) muscle. CM and PM are transcriptomically similar to one another, and TR and MR are also more similar to one another than to CM and PM. The authors then look closer into the expression of effector molecules in the four muscle cell types and find that CM and PM use a similar set of effector molecules that is different from the one of TR/MR muscle, as expected also from the UMAP plot of Figure 1. They then associate the different effector molecules with contraction speed and show that fast contracting muscles are postsynaptic to neurons.

The authors then compare transcription factor expression in TR and MR and show that distinct regulatory molecules regulate the expression of the similar effector molecule signature of the fast-contracting muscles. Afterwards, they compare the regulatory signatures of both fast-contracting muscle cell types with CM and PM and show that the endodermally derived MR muscle shares its regulatory signature with the slow-contracting muscles. They then argue that the diversification of the muscle cell types happened independently in Cnidaria and is tied to the evolution of the bHLH genes *NvNem-7*, *-24*, and *-64*.

The paper touches upon a number of interesting observations and hypotheses, but remains fairly superficial in answering them. While it is a very interesting subject, the authors would need to improve the manuscript significantly to make it more rigorous and accessible to non-*Nematostella* aficionados.

Main comments:

- The annotation of the clusters as CM, PM, TR, and MR is not described in detail. While we have no reason to doubt the authors, they should do a much better job describing how these clusters were annotated, especially if they want to target a broader audience. Figure S1.3B,C are very difficult to read and the gene names are only meaningful to aficionados. The authors have a lot of in situ hybridizations in Figure 2 that probably verify their clusterings, but they should be better presented and made accessible.
- Figure 2B is only accessible to *Nematostella* aficionados. A schematic or annotation of the ISH would significantly improve the presentation of these data.

- The association of effector molecules with contraction speed is handwavy. Could the authors mutate some of the effector molecules and assess this hypothesis? Is anything known about which of the effector genes could be responsible for fast vs slow contraction?
- In Figure 3, the authors argue that the regulatory signatures are shared between MR, PM, and CM, while the effector signature is shared between PM/CM, and TR/MR. It would be nice to see some direct comparison between the entire transcriptome and only the regulatory genes. For example, the authors could do hierarchical clustering of the muscle transcriptomes and then another hierarchical clustering of the transcription factor expression in the muscle cell types. One would expect to see two different clustering, as illustrated in Figure 4A.
- In Figure 3A, all the upper part of the Figure that includes the non-muscle clusters is probably obsolete. Moreover, dot plots, while visually pleasing, can be misleading. The authors should provide some other visualization, e.g. violin plots, (at least as Supplementary material) that includes some kind of statistical comparison of gene expression between different clusters.
- The paragraph that argues that cnidarian muscle diversification is linked to bHLH gene expansion is a handwavy one with little experimental validation. Since it has an interesting hypothesis, I would suggest to move this part to the Discussion and Figures 3C,D into Figure 4, alongside the models of Figure 4B,C.

Minor comments:

- The bulk transcriptomes do not appear to be necessary at any point and are probably redundant. They are not specific enough to annotate cell types (they compare mesentery non-muscle with mesentery retractor muscle) in Figure 1 and they do not offer anything in the next Figures either.
- ImM is not defined in Figure 1.
- Why did the authors completely ignore the longitudinal endodermal muscles? They are only briefly mentioned in the Introduction.
- The authors refer to a core regulatory complex of Nk2, Tbx1/10, Tbx20, HAND and GATA in line 228. As they mention in the abstract, the CoRC describes a collection of *physically interacting* transcription factors. Do the authors have any indication or experimental proof that the molecules mentioned above are physically interacting? If not, they should probably avoid using the term CoRC here.
- The use of distinct regulators to achieve the same phenotype is not something new and is, in fact, pretty well established in worm and fly nervous systems, as elegantly shown over the years by the Hobert lab. I believe that the authors should refer to this work in the Discussion (e.g. around line 243) (reviewed in PMID: 30300603)

Reviewer #2 (Remarks to the Author):

This manuscript seeks to explore the evolutionary relationship of different muscle cell types within *Nematostella* and between *Nematostella* and Bilateria. The authors cite Arendt et al., 2016 as a model for re-constructing cell types, which asserts that cell type identity is defined by their core regulatory complex (CoRC), rather than their phenotype (structure and function) and that cell types evolve through changes in the CoRC. The authors should be applauded for attempting to discuss the evolution of cell types within this theoretical framework, which is more evolutionarily sound than the more commonly applied approach of discussing cell type evolution by overall similarity in transcriptional profile. That being said, the authors often transgress from this model when discussing overall similarity in structural genes, similarity in PSD and expression patterns. This paper would be better presented by stating what the CoRC's of each cell type are in vertebrates and cnidarians, and testing hypotheses of evolutionary relationships by showing the explicit changes/similarities of the CoRC's between cell types. Instead, their somewhat complex results are shoehorned into a neat evolutionary story that often contradicts the data and the evolutionary model.

The data presented is extensive and sound. The clustering of the different muscle cell types in *Nematostella* is interesting, but not unexpected, given that they have distinct functions and morphologies. The TEM work and the CRISPR knock outs also provide additional support for their descriptions of the different cell types, but again are not unexpected based on their data and previous work. The TEM confirms the description of NMJs in retractor muscles and the CRISPR confirms that the gene specific to tentacle retractor muscle affects tentacle retraction. A more compelling CRISPR experiment (albeit more difficult) would be to put some of the paralogs of structural genes under the control of different promoters to determine if they effect cell type function.

My specific suggestions are below:

- The sentence starting at line 32 of the abstract is not coherent without reading the entire paper. Perhaps a more general sentence to say that the two fast contraction muscle cell types have distinct CoRC's but similar effector modules, suggesting a co-option of effector modules from evolutionarily distinct cell types.
- Line 119 states that the variants in structural proteins likely convey properties of force and speed. However, couldn't their NMJ also effect speed? Without functional data, I would suggest that this sentence be deleted.
- Figure 1a is confusing and somewhat misleading. Why show a tree if the goal is to reconstruct the relationships between cell types? Also, what are the purple, green and gray dots supposed to symbolize? Instead, simply listing the types of muscle cell in Bilateria and Cnidaria, their CoRC's and their germ layer origin would be clearer.
- Line 141 state that "...despite their different developmental origins..." the authors should state explicitly what they mean. Different germ layers? Different CoRC's? Arendt et al., 2016 was very explicit in stating that developmental origins are not good evidence for homology so this is not really relevant.

- The authors report extensive gene duplication and subfunctionalization for muscle-specific genes. For each paralogous group, it would be important to see a gene tree to determine if closer paralogs share similar expression profiles, as compared to more distant ones.
- Figure 4a. The tree showing neuronal cells related to tentacle retractor muscles is one of the most interesting findings of this manuscript, but is buried in one sentence and a small detail in a figure. I would suggest expanding on this finding.
- The colored circles in Figure 4c are confusing. For example, it shows that the green ancestral slow muscle transforms into purple fast somatic muscle and then re-transforms into smooth muscle – which is not what the data supports. And the curved arrow pointing to ectodermal *Nematostella* Fast Muscle implies it evolved from *Nematostella* slow muscle cells.
- Not all bilaterians have striated cardiac muscle. Please be specific and instead state vertebrates.
- Also, conclusions of close evolution origins between cardiac muscle and cnidarian slow retracting muscles should be better illustrated in specifically stating the similarities in CoRC and drawing the tree such that they are their closest relatives (meaning - draw a cell type tree, not a species tree with a complex set of characters listed at the terminals).

In summary, although the data presented is solid and extensive, the interpretation of the data is somewhat confusing, as well as their test of the CoRC model of evolution. In addition, as discussed above some of diagrams in the figures are inaccurate, confusing and/or misleading. Although the conclusions that are supported are a significant contribution to the *Nematostella* community, without a more explicit hypothesis testing framework for the evolution of these cell types, this contribution is likely not of broad general interest outside of this specific set of researchers.

Reviewer #3 (Remarks to the Author):

This manuscript by Cole et al. presents a wide-ranging study of muscle evolution using the model cnidarian *Nematostella vectensis*. Cnidaria is the sister taxon to bilaterians and it is thus well positioned to provide insight into the evolution of the canonical muscles seen in bilaterians. This work extends previous work from some of the authors on the evolution of striated muscle in metazoans.

The authors take advantage of the ability to make transgenic *Nematostella* to generate strains from which tissue-specific cells can be sorted. They also carry out scRNAseq on dissected tissues. The resulting data fit nicely with the muscle types already known from anatomical studies of the *Nematostella* polyp, both confirming the anatomical studies and validating the techniques.

A major point the authors make is that gene duplication has played a major role in muscle type diversification. While this is an important finding, I would be interested to know if this is a standard feature of evolution of tissues in animals, or is muscle tissue distinct in this regard?

In addition to the extensive gene content and expression data, the authors make contraction speed measurements on the muscles of the polyp and carry out EM studies of nerve cell/muscle cell junctions. The latter are particularly interesting. What constitutes a synapse in cnidarians has been a puzzling question since Jane Westfall's EM studies on nerve cells in Hydra. The authors experiments using a Homer-mCherry transgenic line to visualize synapses are a major step forward in answering this question. Do the number and distribution of the Homer-positive structures make physiological sense? i.e. does it seem like there are enough synapses to operate the musculature in an animal the size of the *Nematostella* polyp? For comparison, how densely distributed are the neuromuscular junctions in other invertebrates, e.g. *Drosophila* and *C. elegans*?

Finally, the authors use CRISPR-cas9 gene editing to examine the roles of e-protein and nem64 in muscle development. The results they obtain are consistent with the expected roles of these two proteins.

In summary, there is much to like about this paper. It presents a large body of high quality work that establishes a broad understanding of muscle formation and function in *Nematostella*. I have only one suggestion for a change in the paper. I feel that the current title is too restrictive and fails to alert the reader to the diversity of studies and findings that the paper presents. I think a title that provides a better description of the contents of the paper is warranted.

Signed,

Rob Steele

Reviewer #4 (Remarks to the Author):

In the manuscript "Muscle cell type diversification is driven by extensive gene duplications" Cole and colleagues use bulk and single cell RNA sequencing and CRISPR gene editing to characterize and test genes involved in the origin and diversification of muscle cells in the sea anemone *Nematostella vectensis*. The sea anemone is a particularly interesting animal to investigate as it is a diploblast, which is the sister group to the triploblastic Bilateria. While anemones develop from two germ layers (endoderm and ectoderm), they do give rise to muscle, which in triploblasts develops from the third mesodermal germ layers. Thus it is of great interest to understand how muscle develops in diploblasts and how this relates to the development of muscle in triploblasts. They identify 4 muscles in *Nematostella*: 2 fast

muscles, TR that is ectoderm-derived and MR that is endoderm-derived and PM and 2 slow muscles, CM and PM that are endoderm-derived. They find that these muscles display a mixture of similarities and differences in their regulatory genes and structural proteins. The two fast muscles – although sharing expression of similar structural genes- display disparate core regulatory transcriptional profiles, reflective of their embryonic origin. They also beautifully demonstrate by CRISPR knock-outs that the NVEprotein is required for development of all Nematostella muscle and NvNem64 is specifically required for the TR, tentacle retractor muscle. Overall, buried in this dense paper are important and interesting data yielding insights into the regulation of muscle development in Nematostella. These findings also have unique insights into the evolution of muscle in triploblasts. However, the significance of the paper is significantly marred by the dense and difficult to follow text that is made more challenging by poorly defined terms and long lists of genes.

1. The title of the manuscript suggests that gene duplications are the main driver of muscle cell type diversification. The vast majority of the text and main figures, however, are devoted to characterizing expression profiles rather than testing the role of gene duplication. Illustration of the extent of gene duplications within cnidarians should be in one of the main figures, rather than just the supplement. While the deletion of Nem64 provides compelling evidence for its requirement in tentacle retractor muscles, to test the role of gene duplication, the putative “hijacked” or duplicated gene, Nem7 or Nem24, should be deleted. It would be expected that deletion of Nem7 or Nem24 genes would lead to loss of the MR muscle. However, such an experiment is only required if the main point of the paper is that muscle diversification is driven by gene duplication. It is not entirely clear to me that this is the main point of the paper.

2. The authors suggest that the presence of neuromuscular junctions is important for defining the similarity between fast TR and MR muscles. They beautifully show Homer mCherry expression in these two muscles. However, it is also important to show that NMJs and Homer are not expressed in CM and PM muscles and therefore is a unique and defining feature of TR and MR muscles.

3. The authors claim that the endodermal muscle regulatory genes are most similar to the bilaterian cardiomyocyte regulatory genes (p.6-7), but nowhere is this data presented in a figure other than the summary Figure 4C panel. This data should be presented graphically in a figure.

4. I found this paper extremely difficult to read. It is essential that the authors find someone outside their lab to read and heavily edit this paper. The authors should decide on the essential main points and provide only the data need to support these main points. They should endeavor to also reduce the large numbers of lists of genes found in the text so that the text is more readable. These lists should only be included in the figures.

5. Figure 4 provides 3 different models. It is not clear that all three models are needed. Could they be collapsed into one model?

6. The authors use the terms core regulatory complex (“CoRV, a collection of physically interacting transcription factors that together specify the terminal phenotype of a cell” p.1 Abstract), regulatory signature/profile/network and effector modules/genes/molecules (undefined). I am unclear the exact definitions of these terms, other than CoRV, which is defined in the Abstract. Please stick with a limited number of terms and define them explicitly in the beginning of the paper.

Response to reviewers

We thank all reviewers for their helpful comments on our manuscript. We have revised the paper accordingly, and added new data regarding our mutant line that we feel further strengthens the manuscript as a whole. Further, in the effort to streamline the manuscript we have removed the EM and HOMER-transgenic data from the current submission.

Reviewer #1 (Remarks to the Author):

In this paper, Cole et al aim to use single-cell sequencing to discern the muscle cell types in the sea anemone *Nematostella*, address the germ layer origin of these cell types, and compare with the bilaterian muscle cell types. They first identify four transcriptomically separable cell types, circular muscle (CM), parietal muscle (PM), tentacle retractor (TR) and mesentery retractor (MR) muscle. CM and PM are transcriptomically similar to one another, and TR and MR are also more similar to one another than to CM and PM. The authors then look closer into the expression of effector molecules in the four muscle cell types and find that CM and PM use a similar set of effector molecules that is different from the one of TR/MR muscle, as expected also from the UMAP plot of Figure 1. They then associate the different effector molecules with contraction speed and show that fast contracting muscles are postsynaptic to neurons.

The authors then compare transcription factor expression in TR and MR and show that distinct regulatory molecules regulate the expression of the similar effector molecule signature of the fast-contracting muscles. Afterwards, they compare the regulatory signatures of both fast-contracting muscle cell types with CM and PM and show that the endodermally derived MR muscle shares its regulatory signature with the slow-contracting muscles. They then argue that the diversification of the muscle cell types happened independently in Cnidaria and is tied to the evolution of the bHLH genes *NvNem-7*, *-24*, and *-64*.

The paper touches upon a number of interesting observations and hypotheses, but remains fairly superficial in answering them. While it is a very interesting subject, the authors would need to improve the manuscript significantly to make it more rigorous and accessible to non-*Nematostella* aficionados.

Main comments:

- The annotation of the clusters as CM, PM, TR, and MR is not described in detail. While we have no reason to doubt the authors, they should do a much better job describing how these clusters were annotated, especially if they want to target a broader audience. Figure S1.3B,C are very difficult to read and the gene names are only meaningful to aficionados. The authors have a lot of in situ hybridizations in Figure 2 that probably verify their clusterings, but they should be better presented and made accessible.

We thank the reviewer for this comment. We have modified how we present this section, and have included a revised supplementary figure (Fig. S1.3) with a more detailed presentation of the muscle cell cluster identification. While gene names remain meaningful only aficionados, we now include also a more general GO-term analysis that we hope will be more accessible to a wider audience. We have also added schematics to the in situs in figure 2 to better highlight the specific muscle territories within the animal. We have also revised the methods section to better reflect our annotation strategy.

- Figure 2B is only accessible to *Nematostella* aficionados. A schematic or annotation of the

ISH would significantly improve the presentation of these data.

To make the ISH more accessible to a wider audience, we added a schematic to the in situ panel in figure 2, as well as a labelled mesentery that corresponds to the schematic in Fig. 1.

- The association of effector molecules with contraction speed is handwavy. Could the authors mutate some of the effector molecules and assess this hypothesis? Is anything known about which of the effector genes could be responsible for fast vs slow contraction?

This is an interesting question, but it is difficult to answer. The contraction is regulated by a large set of effector proteins and it could be any combination of these proteins which conveys the higher versus slower contraction speed. Given that the generation of homozygous mutants takes 1.5-2 years in *Nematostella*, we are afraid we have to keep it at the level of a correlation.

- In Figure 3, the authors argue that the regulatory signatures are shared between MR, PM, and CM, while the effector signature is shared between PM/CM, and TR/MR. It would be nice to see some direct comparison between the entire transcriptome and only the regulatory genes. For example, the authors could do hierarchical clustering of the muscle transcriptomes and then another hierarchical clustering of the transcription factor expression in the muscle cell types. One would expect to see two different clustering, as illustrated in Figure 4A.

We thank the reviewer for this suggestion. As Figure 3 was split into Fig. 3 and 4, we have revised the new Figure 5 and this now includes hierarchical clustering using structural vs. regulatory proteins.

- In Figure 3A, all the upper part of the Figure that includes the non-muscle clusters is probably obsolete.

We disagree with the reviewer on this point; The upper part of the figure shows a) the endo(meso)dermally derived muscles share signatures unique to this germ layer, whereas b) the TR shares many TFs with the neural cluster, and c) this also illustrates the point that the muscle candidates are indeed more highly expressed elsewhere. These are all important observations in the manuscript and thus we feel they merit representation in the main figure.

Moreover, dot plots, while visually pleasing, can be misleading. The authors should provide some other visualization, e.g. violin plots, (at least as Supplementary material) that includes some kind of statistical comparison of gene expression between different clusters.

The point of the reviewer is well taken but the statistical information is already supplied in the gene lists (Extended Data S1): fold change, p-values AND adjusted p-value. We therefore don't see the added value of a violin plot in the supplement.

- The paragraph that argues that cnidarian muscle diversification is linked to bHLH gene expansion is a handwavy one with little experimental validation. Since it has an interesting hypothesis, I would suggest to move this part to the Discussion and Figures 3C,D into Figure 4, alongside the models of Figure 4B,C.

Following the advice of the reviewer, we now updated the section heading to remove this hypothesis, and present more clearly our interpretation of the data in terms of the evolution of cnidarian muscle within the final discussion paragraph.

Minor comments:

- The bulk transcriptomes do not appear to be necessary at any point and are probably redundant. They are not specific enough to annotate cell types (they compare mesentery non-

muscle with mesentery retractor muscle) in Figure 1 and they do not offer anything in the next Figures either.

The bulk RNAseq of muscle versus non-muscle mesentery tissue was generated by a different dissociation method and provides a much deeper sequencing as well as a differential gene expression analysis. Moreover, the bulk RNAseq is only not specific with respect to the non-muscle cells, but very specific with respect to the mesentery retractor muscle cells, which this paper is concerned about. We think that this data is an important independent confirmation of the single cell datasets and we therefore prefer to keep them in. The combinatorial power of bulk+scRNASeq is also highlighted in Kratsios & Hobert, 2018.

- ImM is not defined in Figure 1.

This has been corrected.

- Why did the authors completely ignore the longitudinal endodermal muscles? They are only briefly mentioned in the Introduction.

We are in fact not ignoring any muscles. The longitudinal endodermal muscles are the parietal muscles (PM) embedded in the body wall and the mesentery retractor muscles (MR). They are mentioned multiple times in the manuscript. The circular muscles of the mesodermal body wall are also an important part of the investigation.

- The authors refer to a core regulatory complex of Nk2, Tbx1/10, Tbx20, HAND and GATA in line 228. As they mention in the abstract, the CoRC describes a collection of *physically interacting* transcription factors. Do the authors have any indication or experimental proof that the molecules mentioned above are physically interacting? If not, they should probably avoid using the term CoRC here.

This is correct, there is at present no indication that these factors do physically interact. We have removed reference to the CoRC model of cell type evolution here.

- The use of distinct regulators to achieve the same phenotype is not something new and is, in fact, pretty well established in worm and fly nervous systems, as elegantly shown over the years by the Hobert lab. I believe that the authors should refer to this work in the Discussion (e.g. around line 243) (reviewed in PMID: 30300603)

We thank the reviewer for pointing out this oversight on our part. The work of Hobert and others is referenced in this context now.

Reviewer #2 (Remarks to the Author):

This manuscript seeks to explore the evolutionary relationship of different muscle cell types within Nematostella and between Nematostella and Bilateria. The authors cite Arendt et al., 2016 as a model for re-constructing cell types, which asserts that cell type identity is defined by their core regulatory complex (CoRC), rather than their phenotype (structure and function) and that cell types evolve through changes in the CoRC. The authors should be applauded for attempting to discuss the evolution of cell types within this theoretical framework, which is more evolutionarily sound than the more commonly applied approach of discussing cell type evolution by overall similarity in transcriptional profile. That being said, the authors often transgress from this model when discussing overall similarity in structural genes, similarity in PSD and expression patterns. This paper would be better presented by stating what the CoRC's of each cell type are in vertebrates and cnidarians, and testing hypotheses of evolutionary relationships by showing the explicit changes/similarities of the CoRC's between cell types.

We appreciate the reviewer's comments in this regard. After much consideration of comments from all reviewers, we have removed discussion of the CoRC model of cell type evolution from the manuscript, as we do not provide data that can be used to test this directly. However we retain our discussion of the use of similar regulatory proteins between vertebrates and cnidarians; it is noteworthy that besides some globally expressed TFs (e.g. e-protein, SRF), the majority of factors is cnidarian or bilaterian specific, with the exception of cardiac/visceral muscle, where we do find a larger set of TFs co-expressed in the parietal muscle.

Instead, if their somewhat complex results are shoehorned into a neat evolutionary story that often contradicts the data and the evolutionary model.

In our revised manuscript, we now focus more on the relationship of the ectodermally-derived tentacle muscle with other muscles in cnidarians in general. We present two alternative interpretations and clarify the support for our preferred interpretation. The focus on the relationships with bilaterian muscle overall is still mentioned, but is a less prominent part of the current paper, as we would need more information from other cnidarians for this.

The data presented is extensive and sound. The clustering of the different muscle cell types in *Nematostella* is interesting, but not unexpected, given that they have distinct functions and morphologies.

We do not fully agree with this statement: The parietal muscle (PM) and the mesentery retractor muscle (MR) are anatomically rather similar, but express distinct set of effector molecules, while MR and tentacle retractor muscle (TR), while being quite distinct at the morphological level, share largely the same set of effector molecules. Prior to generating the data in this study, we would have expected the PM and RM to be most similar in molecular composition.

The TEM work and the CRISPR knock outs also provide additional support for their descriptions of the different cell types, but again are not unexpected based on their data and previous work. The TEM confirms the description of NMJs in retractor muscles and the CRISPR confirms that the gene specific to tentacle retractor muscle affects tentacle retraction. A more compelling CRISPR experiment (albeit more difficult) would be to put some of the paralogs of structural genes under the control of different promoters to determine if they effect cell type function.

This is an interesting suggestion and we would love to do it. Technically it is however challenging on multiple levels and we therefore have to refrain from it for this paper: it is not clear, which of the dozen paralogs of structural protein genes to choose for this experiments, which promoter region would be sufficient to drive the expression. Furthermore, the proper experiment would be to exchange the corresponding paralog using a knockin. Such experiments have now been done in isolated cases but they remain very challenging.

My specific suggestions are below:

- The sentence starting at line 32 of the abstract is not coherent without reading the entire paper. Perhaps a more general sentence to say that t two fast contraction muscle cell types have distinct CoRC's but similar effector modules, suggesting a co-option of effector modules from evolutionarily distinct cell types.

We have re-structured the abstract for clarity.

- Line 119 states that the variants in structural proteins likely convey properties of force and speed. However, couldn't their NMJ also effect speed? Without functional data, I would suggest that this sentence be deleted.

True, the NMJs and neuronal activation could also affect speed. It is, however, known from vertebrates that at least variants of MyHC affect the contraction speed in the same cells as referenced in our manuscript.

- Figure 1a is confusing and somewhat misleading. Why show a tree if the goal is to reconstruct the relationships between cell types? Also, what are the purple, green and gray dots supposed to symbolize? Instead, simply listing the types of muscle cell in Bilateria and Cnidaria, their CoRC's and their germ layer origin would be clearer.

Our goal in the first figure is to illustrate the differences in the germ layer origins of muscles in bilaterians and cnidarians as an introduction into the topic, as well as summarize what is known about the cell types in both sublineages. As we are not truly mapping characters for the purpose of ancestral state reconstruction in this figure, we have modified it slightly to no longer represent a 'tree'. The "CoRC" on the other hand, is a highly specialized term that is familiar to only a sub-set of the readership. There is ample evidence in the literature (summarized and expanded in Brunet et al) that there were 2 ancestral muscle programs corresponding to fast/skeletal (purple) and slow/visceral (green) in bilaterians. One of the key outstanding questions is the relationship between similar fast and slow contracting muscle types in Cnidarians. The cnidarian cell types here are represented as grey because prior to the current manuscript whether these are 'fast' or 'slow' has been unexplored. We then use the purple and green colour scheme to represent these two categories throughout the rest of the manuscript. We have updated the figure caption to better describe the figure.

- Line 141 state that "...despite their different developmental origins..." the authors should state explicitly what they mean. Different germ layers? Different CoRC's? Arendt et al., 2016 was very explicit in stating that developmental origins are not good evidence for homology so this is not really relevant.

The text has been modified and this phrase no longer appears in the manuscript. We believe that the hypothesis of Arendt et al 2016 that the developmental origin does not matter for homology is not yet widely accepted in the community. We therefore feel we should emphasize this point.

- The authors report extensive gene duplication and subfunctionalization for muscle-specific genes. For each paralagous group, it would be important to see a gene tree to determine if closer paralogs share similar expression profiles, as compared to more distant ones.

This is an important point indeed and we had provided phylogenetic trees in Figure. S2.3 for several structural proteins with their *Nematostella* muscle expression highlighted. The trees show that paralogs have been recruited independently to slow and fast muscles, both in vertebrates as well as in cnidarians; These data are now summarized also in the main figure 2e. Additionally we also provide in Supplemental Figure S3.2 the phylogenetic analysis of the bHLH transcription factors, showing the sea anemone specific diversification of bHLH proteins and their use in specific muscles.

- Figure 4a. The tree showing neuronal cells related to tentacle retractor muscles is one of the most interesting findings of this manuscript, but is buried in one sentence and a small detail in a figure. I would suggest expanding on this finding.

We agree with reviewer on the significance of this finding and we expand on this point in the

current version of the paper. Within the results section (line 156-157) we expand upon the common neural markers observed and added double insitu results demonstrating the co-expression of the nem64 muscle marker and a marker of neuronal progenitors to figure 3, as well as expression of a metabotropic glutamate receptor to figure S3.1e. Within the discussion we further comment on the possible evolutionary scenario where nem64 expression within the neurectoderm leads to the cooption of the muscle module and generation of a novel cell type within the ectoderm (lines 243:249).

- The colored circles in Figure 4c are confusing. For example, it shows that the green ancestral slow muscle transforms into purple fast somatic muscle and then re-transforms into smooth muscle – which is not what the data supports. And the curved arrow pointing to ectodermal Nematostella Fast Muscle implies it evolved from Nematostella slow muscle cells. Figure 4c is no longer included in the manuscript.

- Not all bilaterians have striated cardiac muscle. Please be specific and instead state vertebrates.

Figure 4c is no longer included in the manuscript.

- Also, conclusions of close evolution origins between cardiac muscle and cnidarian slow retracting muscles should be better illustrated in specifically stating the similarities in CoRC and drawing the tree such that they are their closest relatives (meaning - draw a cell type tree, not a species tree with a complex set of characters listed at the terminals).

In our revised manuscript we do not emphasize these similarities and as such have also removed Fig. 4c; the hypothesis that the bilaterian cardiac gene set is ancestral and already present in Cnidarians has been proposed previously (Wijesena et al 2017; Steinmetz et al 2017), and so our data contribute added support of this proposed ancestral relationship, and demonstrates that this regulatory module is used not only in early development but also in the formation of differentiated muscle cells. We wish to highlight here rather the putative origin of a novel cell type, neuro-ectodermally derived TR.

In summary, although the data presented is solid and extensive, the interpretation of the data is somewhat confusing, as well as their test of the CoRC model of evolution. In addition, as discussed above some of diagrams in the figures are inaccurate, confusing and/or misleading. Although the conclusions that are supported are a significant contribution to the Nematostella community, without a more explicit hypothesis testing framework for the evolution of these cell types, this contribution is likely not of broad general interest outside of this specific set of researchers.

We thank the reviewer for several suggestions how to improve the manuscript in order to convey the important messages in a more transparent and convincing manner. We should emphasize that this is the first molecular dissection of the muscle cell types in a diploblastic organism, revealing not only an interesting diversity, but also offering a molecular mechanism that has led to this diversification of molecular and physiological features of the muscle cell types.

Reviewer #3 (Remarks to the Author):

This manuscript by Cole et al. presents a wide-ranging study of muscle evolution using the model cnidarian *Nematostella vectensis*. Cnidaria is the sister taxon to bilaterians and it is thus well positioned to provide insight into the evolution of the canonical muscles seen in bilaterians. This work extends previous work from some of the authors on the evolution of

striated muscle in metazoans.

The authors take advantage of the ability to make transgenic *Nematostella* to generate strains from which tissue-specific cells can be sorted. They also carry out scRNAseq on dissected tissues. The resulting data fit nicely with the muscle types already known from anatomical studies of the *Nematostella* polyp, both confirming the anatomical studies and validating the techniques.

A major point the authors make is that gene duplication has played a major role in muscle type diversification. While this is an important finding, I would be interested to know if this is a standard feature of evolution of tissues in animals, or is muscle tissue distinct in this regard? This is indeed an interesting question. As mentioned in the discussion, so far, we only found few examples in the literature from paralogous myosins that define specific subclasses of muscles in vertebrates. However, data from other systems are only emerging now. Future analyses of single cell data from divergent tissue types should concentrate not only on homologous transcription factors but also on the effect of paralogs on convergence.

In addition to the extensive gene content and expression data, the authors make contraction speed measurements on the muscles of the polyp and carry out EM studies of nerve cell/muscle cell junctions. The latter are particularly interesting. What constitutes a synapse in cnidarians has been a puzzling question since Jane Westfall's EM studies on nerve cells in *Hydra*. The authors experiments using a Homer-mCherry transgenic line to visualize synapses are a major step forward in answering this question. Do the number and distribution of the Homer-positive structures make physiological sense? i.e. does it seem like there are enough synapses to operate the musculature in an animal the size of the *Nematostella* polyp? For comparison, how densely distributed are the neuromuscular junctions in other invertebrates, e.g. *Drosophila* and *C. elegans*?

In response to also the other reviewers, we have re-focused the manuscript towards cell type evolution and have removed the data in question. We appreciate Dr. Steele's comments in this regard and will be sure to incorporate this in a separate work focusing primarily on the neural regulation of the musculature.

Finally, the authors use CRISPR-cas9 gene editing to examine the roles of e-protein and nem64 in muscle development. The results they obtain are consistent with the expected roles of these two proteins.

In summary, there is much to like about this paper. It presents a large body of high quality work that establishes a broad understanding of muscle formation and function in *Nematostella*. I have only one suggestion for a change in the paper. I feel that the current title is too restrictive and fails to alert the reader to the diversity of studies and findings that the paper presents. I think a title that provides a better description of the contents of the paper is warranted.

Since we decided to focus on the role bHLH and effector gene duplications we changed the title slightly to:

Muscle cell-type diversification driven by bHLH transcription factor expansion and extensive effector gene duplications

Signed,
Rob Steele

Reviewer #4 (Remarks to the Author):

In the manuscript “Muscle cell type diversification is driven by extensive gene duplications” Cole and colleagues use bulk and single cell RNA sequencing and CRISPR gene editing to characterize and test genes involved in the origin and diversification of muscle cells in the sea anemone *Nematostella vectensis*. The sea anemone is a particularly interesting animal to investigate as it is a diploblast, which is the sister group to the triploblastic Bilateria. While anemones develop from two germ layers (endoderm and ectoderm), they do give rise to muscle, which in triploblasts develops from the third mesodermal germ layers. Thus it is of great interest to understand how muscle develops in diploblasts and how this relates to the development of muscle in triploblasts. They identify 4 muscles in *Nematostella*: 2 fast muscles, TR that is ectoderm-derived and MR that is endoderm-derived and PM and 2 slow muscles, CM and PM that are endoderm-derived. They find that these muscles display a mixture of similarities and differences in their regulatory genes and structural proteins. The two fast muscles – although sharing expression of similar structural genes- display disparate core regulatory transcriptional profiles, reflective of their embryonic origin. They also beautifully demonstrate by CRISPR knock-outs that the NvEprotein is required for development of all *Nematostella* muscle and NvNem64 is specifically required for the TR, tentacle retractor muscle. Overall, buried in this dense paper are important and interesting data yielding insights into the regulation of muscle development in *Nematostella*. These findings also have unique insights into the evolution of muscle in triploblasts. However, the significance of the paper is significantly marred by the dense and difficult to follow text that is made more challenging by poorly defined terms and long lists of genes.

1. The title of the manuscript suggests that gene duplications are the main driver of muscle cell type diversification. The vast majority of the text and main figures, however, are devoted to characterizing expression profiles rather than testing the role of gene duplication. Illustration of the extent of gene duplications within cnidarians should be in one of the main figures, rather than just the supplement.

We agree with the reviewer that the illustration of gene duplications is important. However, the corresponding gene trees take a lot of space and therefore can only be placed in the supplement. While the full gene trees are still in the supplement, we have added a few summary trees to Figure 2 to further illustrate this point.

While the deletion of Nem64 provides compelling evidence for its requirement in tentacle retractor muscles, to test the role of gene duplication, the putative “hijacked” or duplicated gene, Nem7 or Nem24, should be deleted. It would be expected that deletion of Nem7 or Nem24 genes would lead to loss of the MR muscle. However, such an experiment is only required if the main point of the paper is that muscle diversification is driven by gene duplication. It is not entirely clear to me that this is the main point of the paper.

The deletion of nem7 and Nem24 would be indeed desirable to show their role in the development of the endodermal retractor muscle, but we are afraid that this experiment is out of reach for us at present. Due to the relatively long generation time of 4-6 months the generation of a single homozygous mutant takes usually 1,5 years. Plus, the co-expression nem7 and nem24 suggests that they might act redundantly which would require double mutants. We also tried to knockdown these genes by injecting shRNAs against single or both genes, however, the muscles only form at primary polyp stage, when the shRNA has already been degraded.

2. The authors suggest that the presence of neuromuscular junctions is important for defining the similarity between fast TR and MR muscles. They beautifully show Homer mCherry expression in these two muscles. However, it is also important to show that NMJs and Homer are not expressed in CM and PM muscles and therefore is a unique and defining feature of TR and MR muscles.

We agree with the reviewer that it would be desirable to check for localized Homer expression in putative NMJs of CM and PM, however at present we lack the respective specific promoters for those cell types. For this reason and also in response to other reviewers, we have re-focused the manuscript towards cell type evolution and have removed the data in question. We appreciate the reviewers' comments in this regard and will be sure to incorporate this in a separate work focusing primarily on the neural regulation of the musculature.

3. The authors claim that the endodermal muscle regulatory genes are most similar to the bilaterian cardiomyocyte regulatory genes (p.6-7), but nowhere is this data presented in a figure other than the summary Figure 4C panel. This data should be presented graphically in a figure.

In our revised manuscript we do not emphasize these similarities and as such have also removed Fig. 4c; the hypothesis that the bilaterian cardiac gene set is ancestral and already present in Cnidarians has been proposed previously (Wijesena et al 2017; Steinmetz et al 2017), and so our data contribute added support of this proposed ancestral relationship, and demonstrates that this regulatory module is used not only in early development but also in the formation of differentiated muscle cells. We find these observations noteworthy in the manuscript, but we rather prefer to highlight here the putative origin of a novel cell type, neuro-ectodermally derived TR in the main figures.

4. I found this paper extremely difficult to read. It is essential that the authors find someone outside their lab to read and heavily edit this paper. The authors should decide on the essential main points and provide only the data need to support these main points. They should endeavor to also reduce the large numbers of lists of genes found in the text so that the text is more readable. These lists should only be included in the figures.

We have thoroughly revised the manuscript and hope that it is more accessible at this point. Specifically, we streamlined the manuscript by removing the part about the NMJs, focusing on the transcription factor and effector gene set and their duplications. We also removed where we deemed less important the lists of genes.

5. Figure 4 provides 3 different models. It is not clear that all three models are needed. Could they be collapsed into one model?

We largely agree with the reviewer. We have split Figure 4 to make it more accessible and reorganized Figure 5 to better reflect our conclusions and data interpretations; 5a is a summary diagram of the data presented in the paper; 5b is now a hierarchical clustering of transcription factors and effector molecules in the cell clusters; 5c represents our working hypothesis of how muscle evolution occurred within *Nematostella*

6. The authors use the terms core regulatory complex ("CoRV, a collection of physically interacting transcription factors that together specify the terminal phenotype of a cell" p.1 Abstract), regulatory signature/profile/network and effector modules/genes/molecules (undefined). I am unclear the exact definitions of these terms, other than CoRV, which is

defined in the Abstract. Please stick with a limited number of terms and define them explicitly in the beginning of the paper.

We appreciate this comment, and have reconsidered how we present our data accordingly.

We now use consistent terminology throughout, and define terms where they first appear in the main text.

REVIEWER COMMENTS

Reviewer #1 (Remarks to the Author):

The authors have reduced the size of the manuscript and have made it more rigorous addressing the reviewer comments. I still think that there are some minor improvements to be made:

- Line 133: "While some candidate transcription (co-)factors commonly associated with muscle formation in Bilateria (e.g. *srf*, *mef2*, and *myocardin*) are detected in all four muscle cells types, they are also detected within non-muscle ectodermal derivatives at equivalent or even higher (e.g. *mef2*) levels (Fig. 3a: Muscle candidates)."

An in situ for some of these factors, notably *mef2*, showing that it is expressed indeed in non-muscle cells would strengthen their point.

- Line 157: "Interestingly, the TR also expresses ELAV, a marker of a large subpopulation of neurons in *Nematostella*, as well as the neuronal transcription factors *soxB2a* (aka *soxB(2)*, *isl*, *otxA*, and *foxL2*, and a metabotropic glutamate receptor."

Similarly, an in situ against *elav* showing expression in the TR would be beneficial.

- Line 167: "This demonstrates that with respect to transcription factors, the fast mes(endo)dermally derived mesentery retractor muscle (MR) shows greater similarity to the slow muscle (PM, CM) than to its ectoderm-derived counterpart in the tentacle (TR)."

As far as I understand from the hierarchical clustering of Figure 5B, this does not seem to be the case. In the regulatory profile tree, MR is closer to TR than to PM and CM.

Reviewer #2 (Remarks to the Author):

This manuscript is overall much improved from the previous iteration. The presented data is clearer and the explanations are more concise. The authors took considerable effort to address the reviewers concerns. That being said, the one strength of the paper was the evolutionary interpretation based on conservation of the Core Regulatory Complex (CoRC). This was a stronger evolutionary framework than one just based on overall similarity of gene expression. Thus the authors fall into the trap of many other single studies, where they conflate overall similarity in gene expression with evolutionary origin. Given the extensive co-option (convergence) in gene expression, overall similarity is NOT evidence of evolutionary origin and thus authors should adjust their language accordingly (outlined below).

The most interesting aspect of this paper is that the authors identified four distinct muscle cell types and identified which of these share effector (functional) genes and which cell types share regulatory (transcriptional factor) genes. This is well supported by the data. The functional (CRISPR) experiments strongly support their interpretations.

Minor suggestions for changes are listed below:

- The title implies more general processes that is not supported by this contributions. Thus they should add "... in the sea anemone *Nematostella*..." to the title.
- Page 1, line 17 – replace evolutionary mechanisms with developmental mechanisms as this paper does not explore evolutionary mechanisms (perhaps, but debatably evolutionary patterns but definitely not mechanisms).

- Page 1, line 49. Typo – replace one of the “from”s with develop.
- Page 8, line 226 – there is little evidence for “share a common ancestry” but instead there is evidence that the regulatory network is evolutionarily conserved between these two cell types in Nematostella and Bilateria.
- Page 9 line 245 – medusazoan is misspelled. It should be medusozoan.
- Page 27 line 649 – Typo - tenatcles should be tentacles
- Page 29 line 637 Typo – anothozan should be anthozoan.
- Page 29 line 644 – c should be labeled b.

Reviewer #3 (Remarks to the Author):

The authors have satisfactorily addressed my comments in the original review.

Signed,
Rob Steele

Reviewer #4 (Remarks to the Author):

The revised manuscript “Muscle cell-type diversification is driven by bHLH transcription factor expansion and extensive effector gene duplications” has undergone a substantial revision in focus and presentation. I found the first version difficult to read, and this new version is much easier to follow. Removal of the section on innervation helps to keep the paper appropriately focused on the genetic origin and diversification of muscle cells in Nematostella. They have largely addressed my previous concerns about the paper. The paper provides important insights into the origin of skeletal muscle in diploblasts and will be of interest to evolutionary and skeletal muscle biologists.

REVIEWER COMMENTS

Reviewer #1 (Remarks to the Author):

The authors have reduced the size of the manuscript and have made it more rigorous addressing the reviewer comments. I still think that there are some minor improvements to be made:

- Line 133: "While some candidate transcription (co-)factors commonly associated with muscle formation in Bilateria (e.g. *srf*, *mef2*, and *myocardin*) are detected in all four muscle cells types, they are also detected within non-muscle ectodermal derivatives at equivalent or even higher (e.g. *mef2*) levels (Fig. 3a: Muscle candidates)."

An in situ for some of these factors, notably *mef2*, showing that it is expressed indeed in non-muscle cells would strengthen their point.

Response: *mef2* expression in non-muscle cells has been well characterized previously, and we added the reference here (line 133: "as has been previously documented (Genikhovich and Technau 2011)).

- Line 157: "Interestingly, the TR also expresses ELAV, a marker of a large subpopulation of neurons in *Nematostella*, as well as the neuronal transcription factors *soxB2a* (aka *soxB(2)*), *isl*, *otxA*, and *foxL2*, and a metabotropic glutamate receptor."

Similarly, an in situ against *elav* showing expression in the TR would be beneficial.

While an in situ for the glutamate receptor is already present within the supplemental material (supplementary figure 8e), we added also here a double fluorescent in situ demonstrating overlapping expression of *nem64* and *elav* within the tentacles (sup. Fig 8f).

- Line 167: "This demonstrates that with respect to transcription factors, the fast mes(endo)dermally derived mesentery retractor muscle (MR) shows greater similarity to the slow muscle (PM, CM) than to its ectoderm-derived counterpart in the tentacle (TR)."

As far as I understand from the hierarchical clustering of Figure 5B, this does not seem to be the case. In the regulatory profile tree, MR is closer to TR than to PM and CM.

We appreciate the concern raised here. The topology of the trees is influenced also by the lack of inclusion of an ectodermal outgroup. We have re-ran this analysis considering the entire Tissue dataset (rather than simply the gastrodermal+ectodermal muscle subset) and have updated figure 5 accordingly.

Reviewer #2 (Remarks to the Author):

This manuscript is overall much improved from the previous iteration. The presented data is clearer and the explanations are more concise. The authors took considerable effort to address the reviewers concerns. That being said, the one strength of the paper was the evolutionary interpretation based on conservation of the Core Regulatory Complex (CoRC). This was a stronger evolutionary framework than one just based on overall similarity of gene expression. Thus the authors fall into the trap of many other single studies, where they conflate overall similarity in gene

expression with evolutionary origin. Given the extensive co-option (convergence) in gene expression, overall similarity is NOT evidence of evolutionary origin and thus authors should adjust their language accordingly (outlined below).

The most interesting aspect of this paper is that the authors identified four distinct muscle cell types and identified which of these share effector (functional) genes and which cell types share regulatory (transcriptional factor) genes. This is well supported by the data. The functional (CRISPR) experiments strongly support their interpretations.

Minor suggestions for changes are listed below:

- The title implies more general processes that is not supported by this contributions. Thus they should add "... in the sea anemone Nematostella..." to the title.

Although we recognise the reviewers concern here, we feel this addition would alienate a great deal of the readership whom would otherwise be interested in our findings. We thus leave this change to the editors' discretion.

- Page 1, line 17 – replace evolutionary mechanisms with developmental mechanisms as this paper does not explore evolutionary mechanisms (perhaps, but debatably evolutionary patterns but definitely not mechanisms).

We dropped 'evolutionary' from the sentence, but do not specify 'developmental' as this is a general interest statement which introduces the theme of the paper, which encompasses both.

- Page 1, line 49. Typo – replace one of the "from"s with develop.

The original text was not a typo: 'form from'. However we changed "form" to "develop" for clarity.

- Page 8, line 226 – there is little evidence for "share a common ancestry" but instead there is evidence that the regulatory network is evolutionarily conserved between these two cell types in Nematostella and Bilateria.

To clarify our intent in this direction, we reversed the order of our presentation of the data and the interpretation as follows:

"While the presence of a bilaterian cardiac regulatory gene network has been previously noted^{9,30}, we demonstrate here that these genes are used not only in germ layer specification but also in the determination of specific muscle sub-types. Thus sea anemone slow muscles are likely to share a common ancestry with bilaterian cardiac/smooth muscle cell type. "

- Page 9 line 245 – medusazoan is misspelled. It should be medusozoan.

Corrected.

- Page 27 line 649 – Typo - tenatcles should be tentacles

Corrected.

- Page 29 line 637 Typo – anothozan should be anthozan.

Corrected.

- Page 29 line 644 – c should be labeled b.

Corrected.

Reviewer #3 (Remarks to the Author):

The authors have satisfactorily addressed my comments in the original review.

Thank you for the endorsement of our work.

Signed,
Rob Steele

Reviewer #4 (Remarks to the Author):

The revised manuscript “Muscle cell-type diversification is driven by bHLH transcription factor expansion and extensive effector gene duplications” has undergone a substantial revision in focus and presentation. I found the first version difficult to read, and this new version is much easier to follow. Removal of the section on innervation helps to keep the paper appropriately focused on the genetic origin and diversification of muscle cells in *Nematostella*. They have largely addressed my previous concerns about the paper. The paper provides important insights into the origin of skeletal muscle in diploblasts and will be of interest to evolutionary and skeletal muscle biologists.

Thank you for the endorsement of our work.

REVIEWERS' COMMENTS

Reviewer #1 (Remarks to the Author):

My comments were addressed.